# MoEEdit: Efficient and Routing-Stable Knowledge Editing for Mixture-of-Experts LLMs

**Yupu Gu**[1]**, Rongzhe Wei**[2]**, Andy Zhu**[2]**, Pan Li**[2]

[1]Tsinghua University, [2]Georgia Institute of Technology

guyp22@mails.tsinghua.edu.cn, {rongzhe.wei, azhu311, panli}@gatech.edu

## Abstract

Knowledge editing (KE) enables precise modifications to factual content in large language models (LLMs). Existing KE methods are largely designed for dense architectures, limiting their applicability to the increasingly prevalent sparse Mixture-of-Experts (MoE) models that underpin modern scalable LLMs. Although MoEs offer strong efficiency and capacity scaling, naively adapting dense-model editors is both computationally costly and prone to routing distribution shifts that undermine stability and consistency. To address these challenges, we introduce MoEEdit, the first routing-stable framework for parameter-modifying knowledge editing in MoE LLMs. Our method reparameterizes expert updates via per-expert null-space projections that keep router inputs invariant and thereby suppress routing shifts. The resulting block-structured optimization is solved efficiently with a block coordinate descent (BCD) solver. Experiments show that MoEEdit attains state-of-the-art efficacy and generalization while preserving high specificity and routing stability, with superior compute and memory efficiency. These results establish a robust foundation for scalable, precise knowledge editing in sparse LLMs and underscore the importance of routing-stable interventions. The code for this work is available at https://github.com/Terence-Gu/MoEEdit. [1]

## 1 Introduction

LLMs can store and retrieve substantial factual knowledge (Petroni et al., 2019; Sun et al., 2024), yet they sometimes produce incorrect or outdated statements. For instance, asserting that the capital of France is Berlin or misreporting the CEO of a major company. Such errors undermine user trust and constrain deployment in knowledge-sensitive applications (Zhang et al., 2024c; Zhong et al., 2023; Wei et al., 2024; 2025a;b). Fully retraining these models or performing broad fine-tuning is computationally prohibitive and can induce catastrophic forgetting of unrelated capabilities (Luo et al., 2025). These limitations motivate knowledge editing, which aims to revise specific facts while preserving the model's general behavior (Meng et al., 2022; 2023; Mitchell et al., 2022a;b).

Most KE methods have been designed for dense Transformer architectures, where all parameters are active for each input (Wang et al., 2024). Broadly, KE falls into two families: *parameter-preserving approaches* leave base weights unchanged and attach auxiliary mechanisms that conditionally override outputs (e.g., SERAC with an external edit memory and routing module (Mitchell et al., 2022b)), and *parameter-modifying approaches* aim to directly update model weights responsible for factual recall. Many methods follow a locate-then-edit paradigm: they identify mediating parameters, often mid-layer feed-forward MLP modules (Geva et al., 2021; Dai et al., 2022), using causal analyses, and then apply structured weight updates. Representative methods include ROME (Meng et al., 2022), MEMIT (Meng et al., 2023), and PMET (Li et al., 2024). Recent work improves locality by projecting edits into the null space of a preservation set, which reduces interference with unrelated behaviors (Fang et al., 2025).

---

[1]Work was done when Yupu Gu and Andy Zhu did research internship at Georgia Tech.

State-of-the-art LLMs increasingly adopt Mixture-of-Experts (MoE) architectures to enlarge parameter capacity while maintaining nearly constant computational throughput (FLOPs) (Shazeer et al., 2017). In an MoE layer, a trainable router activates a small subset of experts for each token (for instance, 8 of 128 in Qwen3-30B-A3B), yielding sparse, input-dependent computation and enabling marked expert specialization (Lepikhin et al., 2021; Du et al., 2022). However, this sparse, modular design introduces a tripartite challenge for knowledge editing that is absent in dense models.

The first and most direct challenge is *computational complexity*. Naively applying dense-model editing techniques would require updating all experts, multiplying the cost by their total number (e.g., $128\times$) and thus rendering the process computationally prohibitive. This necessitates a targeted approach, but editing only a subset of experts introduces further complications. Second, because the layer's output is a gate-weighted combination of multiple expert outputs, any edit must contend with *inter-expert coupling*. A modification to a single expert's parameters can be diluted or cause unintended side effects when combined with others, demanding a principled allocation of the update across the appropriate specialists. Third, and most subtly, edits risk inducing *routing distribution shifts* in subsequent layers. Parameter perturbations in one MoE layer alter the input manifold for downstream layers, causing their routers to select different experts. This cascading effect disrupts the model's learned routing patterns and specialized knowledge pathways, jeopardizing both edit locality and overall model stability. Collectively, these intertwined issues of computational cost, expert coupling, and routing stability make successful and localized knowledge editing in MoEs substantially more difficult than in their dense counterparts.

To address this tripartite challenge, we introduce MoEEdit, an expert-aware editor that reframes MoE knowledge editing as a principled, block-structured optimization problem where each expert constitutes a block. We are the first to formally identify routing-induced instability as a central obstacle to successful editing in MoEs. To solve this, we develop a novel *per-expert null-space projection* that constrains parameter updates to preserve the input to subsequent routers, thereby safeguarding model stability. This technique is paired with a highly efficient *randomized block coordinate descent (BCD) solver* that tackles computational complexity and inter-expert coupling by strategically updating only the most relevant experts for a given edit. This decoupling ensures our method's cost scales linearly with the expert hidden size, not the total number of experts, making it highly scalable. Our integrated approach sets a new state-of-the-art on standard benchmarks (COUNTERFACT, zsRE), decisively outperforming dense-model editors adapted for MoEs. This result underscores the necessity of expert-aware, routing-stable interventions tailored to the unique architectural properties of MoE models.

## 2 RELATED WORK

**KE in dense Transformers.** KE seeks to revise specific factual associations in LLMs while preserving general capabilities (Zhang et al., 2024c; Zhong et al., 2023). Methods for dense Transformers fall into two families: *parameter-modifying* and *parameter-preserving*. Within the former, locate-then-edit approaches such as ROME (Meng et al., 2022) and MEMIT (Meng et al., 2023) directly modify a small set of FFN down-projection weights identified via causal analysis, enabling single-fact and batched edits, respectively. Gradient-based editors such as MEND (Mitchell et al., 2022a) learn hypernetworks that transform fine-tuning gradients into localized weight updates. To improve locality and robustness under sequential edits, AlphaEdit (Fang et al., 2025) projects updates into the null space of a preservation set. Other strands explore instruction-based editing (Zhang et al., 2024a), hybrid neural–symbolic mechanisms (Zhang et al., 2024b), and in-context editing (Zheng et al., 2023). Complementing these, *parameter-preserving* or semi-parametric methods (e.g., SERAC (Mitchell et al., 2022b)) store edits in external memories and perform inference-time routing, trading parametric locality for reversibility and capacity. Notably, LEMoE (Wang & Li, 2024) introduces a Mixture-of-Experts (MoE) architecture within the adaptor itself to manage lifelong editing. However, LEMoE functions as a parameter-preserving framework that attaches external modules to a frozen backbone (typically dense). It addresses routing consistency within the added adaptor rather than the routing distribution shift of the base model itself.

**MoE architectures.** MoE layers scale capacity by activating only a sparse subset of experts per token through a learned router, thereby increasing representational power while keeping FLOPs nearly constant (Shazeer et al., 2017; Lepikhin et al., 2021; Du et al., 2022). Each expert is a gated

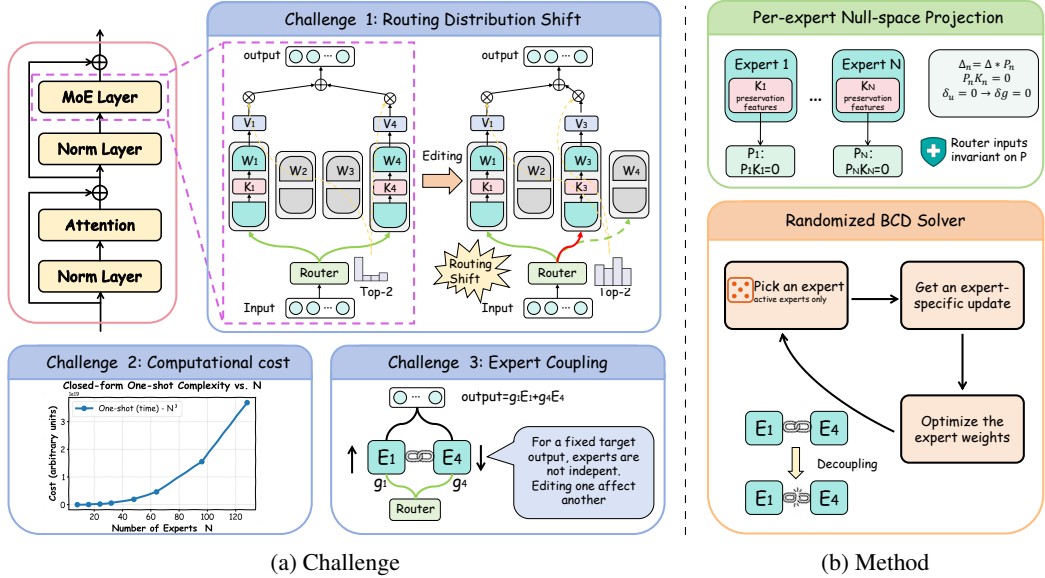

Figure 1: Overview of knowledge editing in Mixture-of-Experts (MoE) LLMs. (a) **Challenges**: MoE editing is hindered by routing distribution shift, high computational cost, and expert coupling. (b) **Method**: MoEEdit mitigates these issues using per-expert null-space projection to stabilize routing and a randomized block coordinate descent (BCD) solver for efficient expert updates.

feed-forward network with its own parameters; the router selects the top-$K$ experts using input-dependent logits and produces a gate-weighted sum of their outputs. This conditional computation yields strong expert specialization but also creates editing complications: edits must respect the router's distribution, multiple experts jointly determine the output, and perturbations at one layer can alter downstream routing.

**Knowledge editing for MoE LLMs.** However, KE for MoE architectures remains largely unexplored. While methods like LEMoE (Wang & Li, 2024) utilize MoE structures externally, they do not tackle the challenge of modifying intrinsic MoE parameters. Existing techniques, which assume fully active parameters, are ill-suited for the conditional computation in MoEs and present an intractable trade-off: updating all experts is computationally prohibitive, while updating a subset is unreliable due to stochastic routing. This leaves a critical gap for an editing framework explicitly designed for the sparse and modular nature of MoE models.

## 3 PRELIMINARIES

**Locate-then-Edit Paradigm (Dense Models).** An autoregressive LLM updates the layer-$l$ hidden state as $\boldsymbol{h}^l = \boldsymbol{h}^{l-1} + \boldsymbol{a}^l + \boldsymbol{v}^l$, where $\boldsymbol{a}^l$ and $\boldsymbol{v}^l$ are the outputs of the attention and feed-forward (FFN) blocks at layer $l$, respectively. The FFN output can be written as

$$\boldsymbol{v}^l = \boldsymbol{W}_{\text{out}}^l \underbrace{\sigma\big(\boldsymbol{W}_{\text{in}}^l \gamma(\boldsymbol{h}^{l-1} + \boldsymbol{a}^l)\big)}_{\text{"keys" } \boldsymbol{k}} = \boldsymbol{W}_{\text{out}}^l \cdot \boldsymbol{k}, \tag{1}$$

with layer norm $\gamma(\cdot)$ and nonlinearity $\sigma(\cdot)$. Following Geva et al. (2021), $\boldsymbol{W}_{\text{out}}^l$ can be viewed as a linear associative memory that maps post-gate features ("keys": $\boldsymbol{k} = \sigma\big(\boldsymbol{W}_{\text{in}}^l \gamma(\boldsymbol{h}^{l-1} + \boldsymbol{a}^l)\big)$) to outputs ("values": $\boldsymbol{v} = \boldsymbol{v}^l$). If factual knowledge is formalized as triples $(s, r, o)$ (subject, relation, object), one can view $\boldsymbol{k}$ as encoding $(s, r)$ and $\boldsymbol{v}$ as encoding $o$ (Meng et al., 2022; Dai et al., 2022). We adopt a locate-then-edit formulation: given keys $\boldsymbol{K}_1 = [\boldsymbol{k}_1 | \boldsymbol{k}_2 | \cdots | \boldsymbol{k}_n]$ for the new associations and targets $\boldsymbol{V}_1 = [\boldsymbol{v}_1 | \boldsymbol{v}_2 | \cdots | \boldsymbol{v}_n]$, find a small perturbation $\boldsymbol{\Delta}$ to a single FFN projection $\boldsymbol{W}_{\text{out}}$ such that $(\boldsymbol{W}_{\text{out}} + \boldsymbol{\Delta})\boldsymbol{K}_1 \approx \boldsymbol{V}_1$ while preserving behavior on a preservation set with keys $\boldsymbol{K}_0$. This yields the regularized least-squares objective

$$\boldsymbol{\Delta} = \arg\min_{\tilde{\boldsymbol{\Delta}}} \big\|(\boldsymbol{W}_{\text{out}} + \tilde{\Delta})\boldsymbol{K}_1 - \boldsymbol{V}_1\big\|^2 + \big\|\tilde{\boldsymbol{\Delta}}\boldsymbol{K}_0\big\|^2 + \lambda\|\tilde{\boldsymbol{\Delta}}\|^2, \tag{2}$$

where $\lambda \geq 0$ controls locality and conditioning. For clarity, layer indices are omitted below since the formulation applies identically to each layer.

**KE in MoE LLMs.** Modern LLMs increasingly adopt MoE layers to scale capacity with near-constant FLOPs (Shazeer et al., 2017; Lepikhin et al., 2021). Given an input representation $\boldsymbol{u}$, a trainable router produces logits $s_n = \boldsymbol{u}^\top \boldsymbol{e}_n$ ($\boldsymbol{e}_n$ is the routing embedding for expert n) and selects a small set $S = \mathrm{TopK}(s_{1:N}, K)$ (TopK selects K biggest element of $s_{1:N}$, and typically $K \ll N$ for MoE models). Let $g_n$ denote the router weight for expert $n$. The MoE block output is a router-weighted mixture

$$\boldsymbol{v} = \sum_{n=1}^{N} g_n\, E_n(\boldsymbol{u}), \qquad g_n = \begin{cases} \exp(s_n) / \sum_{j \in S} \exp(s_j), & n \in S, \\ 0, & \text{otherwise,} \end{cases} \tag{3}$$

where each expert is a gated FFN

$$E_n(\boldsymbol{u}) = \boldsymbol{W}_{\mathrm{down}}^{(n)}\left[\left(\boldsymbol{W}_{\mathrm{up}}^{(n)} \boldsymbol{u}\right) \odot \sigma\left(\boldsymbol{W}_{\mathrm{gate}}^{(n)} \boldsymbol{u}\right)\right]. \tag{4}$$

Thus, every expert realizes its own key–value memory: the post-gate feature $\boldsymbol{k}_n = \left(\boldsymbol{W}_{\mathrm{up}}^{(n)} \boldsymbol{u}\right) \odot \sigma\left(\boldsymbol{W}_{\mathrm{gate}}^{(n)} \boldsymbol{u}\right)$ serves as a key and the linear map $\boldsymbol{W}_{\mathrm{down}}^{(n)}$ returns a value $\boldsymbol{v}_n = \boldsymbol{W}_{\mathrm{down}}^{(n)} \boldsymbol{k}_n$, which the router aggregates into $\boldsymbol{v} = \sum_n g_n \boldsymbol{v}_n$.

Extending Eqn. 2 to MoE, we edit expert-specific projections $\{\boldsymbol{W}_n\}_{n=1}^{N}$ (For brevity in the editing objective, we denote $\boldsymbol{W}_n \equiv \boldsymbol{W}_{\mathrm{down}}^{(n)}$). For edit request $i$, let $\boldsymbol{k}_{i,n} \in \mathbb{R}^{d_k}$ be the post-gate key of expert $n$, $g_{i,n} \geq 0$ its router weight, and $\boldsymbol{v}_i$ the target output. We consider disjoint sets: an edit set $\mathcal{E}$ where outputs should change and a preservation set $\mathcal{P}$ to remain stable. The MoE KE objective seeks small perturbations $\{\boldsymbol{\Delta}_n\}$ that (i) match targets on $\mathcal{E}$ and (ii) preserve behavior on $\mathcal{P}$:

$$\{\boldsymbol{\Delta}_n\} = \arg\min_{\{\tilde{\boldsymbol{\Delta}}_n\}} \sum_{i \in \mathcal{E}} \left\| \sum_{n=1}^{N} g_{i,n}(\boldsymbol{W}_n + \tilde{\boldsymbol{\Delta}}_n)\boldsymbol{k}_{i,n} - \boldsymbol{v}_i \right\|^2 + \sum_{i \in \mathcal{P}} \left\| \sum_{n=1}^{N} g_{i,n} \tilde{\boldsymbol{\Delta}}_n \boldsymbol{k}_{i,n} \right\|^2 + \lambda \sum_{n=1}^{N} \|\tilde{\boldsymbol{\Delta}}_n\|^2. \tag{5}$$

Compared with the dense case, the router weights $\{g_{i,n}\}$ couple experts: each example influences up to $K$ experts (Top-$K$ gating), and a naive closed-form solve would require inverting a $(Nd_k) \times (Nd_k)$ system, computationally prohibitive for large $N$. This coupling and scale motivate the expert-aware optimization strategy developed in Section 4.

## 4 METHOD

In this section, we present MoEEdit, a purpose-built, expert-aware knowledge editor for MoE models. Our approach tackles the unique challenges of MoE editing head-on: (i) it mitigates the routing distribution shift: a key instability when naively apply KE to MoE models by reparameterizing expert updates through per-expert null-space projections , and (ii) solves the resulting objective efficiently with a randomized block coordinate descent (BCD) procedure that scales with the expert hidden size, making large-scale MoE editing tractable.

### 4.1 ROUTING DISTRIBUTION SHIFT IN MOE EDITING

Editing expert parameters changes the MoE block outputs and, after subsequent normalization and attention, perturbs the input $\boldsymbol{u}$ to the router in the following MoE layers. Following the definition in Section 3, let $\boldsymbol{E}_\ell = [\boldsymbol{e}_1^\ell, \ldots, \boldsymbol{e}_N^\ell]$ collect the router embeddings at layer $\ell$. The router computes logits and mixture weights as $\boldsymbol{g}_\ell = \mathrm{softmax}(\boldsymbol{s}_\ell)^2$, where $\boldsymbol{s}_\ell = \boldsymbol{E}_\ell^\top \boldsymbol{u}_\ell$. A perturbation applied in layer $(\ell - 1)$ changes $\boldsymbol{u}_\ell$ by $\delta \boldsymbol{u}_\ell$, thus $\boldsymbol{s}_\ell$ by $\delta \boldsymbol{s}_\ell = E_\ell^\top \delta \boldsymbol{u}_\ell$ and produces new routing weights $\boldsymbol{g}_\ell' = \mathrm{softmax}(\boldsymbol{s}_\ell + \delta \boldsymbol{s}_\ell)$. We define the routing distribution shift as $\delta \boldsymbol{g}_\ell = \boldsymbol{g}_\ell' - \boldsymbol{g}_\ell$. Its magnitude can be quantified over a set of prompts using the Kullback-Leibler (KL) divergence or the Routing

---

[2] For clarity of analysis, we use the full softmax distribution and omit the Top-$K$ selection, so that $\boldsymbol{g}_\ell$ remains differentiable for the Jacobian-based first-order analysis.

Similarity (RS), which is defined as the Jaccard similarity between the pre- and post-edit Top-$K$ expert sets:

$$\text{RS}_{\text{route}}^{\ell} = \frac{|S_{\ell}^{\text{pre}} \cap S_{\ell}^{\text{post}}|}{|S_{\ell}^{\text{pre}} \cup S_{\ell}^{\text{post}}|} \quad \text{or} \quad \text{KL}(\boldsymbol{g}_{\ell} \| \boldsymbol{g}_{\ell}'), \tag{6}$$

where $S_{\ell}^{\text{pre}}$ and $S_{\ell}^{\text{post}}$ denote the expert sets before and after editing, respectively. To characterize $\delta \boldsymbol{g}_{\ell}$ analytically, we linearize the softmax around $\boldsymbol{s}_{\ell}$:

$$\boldsymbol{g}_{\ell}' \approx \boldsymbol{g}_{\ell} + J_{\text{sm}}(\boldsymbol{s}_{\ell}) \delta \boldsymbol{s}_{\ell} \quad \Rightarrow \quad \delta \boldsymbol{g}_{\ell} \approx J_{\text{sm}}(\boldsymbol{s}_{\ell}) \boldsymbol{E}_{\ell}^{\top} \delta \boldsymbol{u}_{\ell}, \tag{7}$$

Where $\approx$ denotes a first-order Taylor approximation of the softmax function around $\boldsymbol{s}_{\ell}$, when the perturbation $\delta \boldsymbol{s}_{\ell}$ is small, and $J_{\text{sm}}(\boldsymbol{s}) = \text{diag}(\text{sm}(\boldsymbol{s})) - \text{sm}(\boldsymbol{s}) \text{sm}(\boldsymbol{s})^{\top}$ is the Jacobian of softmax and sm is the softmax function. This relation highlights a crucial observation: only the component of $\delta \boldsymbol{u}_{\ell}$ that lies in the span of $\boldsymbol{E}_{\ell}$ influences the routing probabilities, and the Jacobian can amplify such components, potentially destabilizing expert selection. This insight motivates our design, suppressing the projection of perturbations onto $\text{span}(\boldsymbol{E}_{\ell})$ is key to preventing routing drift.

## 4.2 PER-EXPERT NULL-SPACE PROJECTION REPARAMETERIZATION

As established in Section 4.1, suppressing routing drift amounts to ensuring that $\delta \boldsymbol{u}_{\ell} \approx \boldsymbol{0}$ on a preservation set. To achieve this by construction, we reparameterize each expert update so that its effect vanishes along directions spanned by preservation features. Inspired by the null-space constrained approach of ALPHAEDIT for dense models, we generalize the idea to the MoE setting by computing a per-expert projector that filters out harmful update components.

Concretely, recall from Eqn. 4 that expert $n$ produces output $\boldsymbol{W}_n \boldsymbol{k}_{i,n}$ for post-activation key $\boldsymbol{k}_{i,n}$. Let $\mathcal{P}$ denote the set of preservation prompts, and collect their features for expert $n$ into the matrix $\boldsymbol{K}_n^0 = \left[ \boldsymbol{k}_{i,n} \right]_{i \in \mathcal{P}} \in \mathbb{R}^{d_k \times |\mathcal{P}|}$. The covariance $\boldsymbol{K}_n^0 \boldsymbol{K}_n^{0\top}$ captures the subspace of activations we wish to keep invariant. We compute its eigendecomposition, $\boldsymbol{K}_n^0 \boldsymbol{K}_n^{0\top} = \boldsymbol{U}_n \Lambda_n \boldsymbol{U}_n^{\top}$, and select indices $\mathcal{I}_0 = \{p : \lambda_{n,p} < \tau\}$ corresponding to (near-)null eigenvalues under a small threshold $\tau > 0$. Let $\boldsymbol{U}_n^0 = \boldsymbol{U}_n[:, \mathcal{I}_0]$ and define the orthogonal projector onto the complement of $\text{span}(\boldsymbol{K}_n^0)$ by $\boldsymbol{P}_n = \boldsymbol{U}_n^0 \boldsymbol{U}_n^{0\top}$. Intuitively, $\boldsymbol{P}_n$ preserves only those directions orthogonal to all preservation features, so any update projected by $\boldsymbol{P}_n$ is guaranteed not to alter expert outputs on $\mathcal{P}$.

We then reparameterize the expert update as $\boldsymbol{\Delta}_n = \hat{\boldsymbol{\Delta}}_n \boldsymbol{P}_n$, where $\hat{\boldsymbol{\Delta}}_n$ is the free variable to be optimized. Because $\boldsymbol{P}_n \boldsymbol{k}_{i,n} = \boldsymbol{0}$ for all $i \in \mathcal{P}$ (up to numerical error), the preservation outputs are unaffected: $\hat{\boldsymbol{\Delta}}_n \boldsymbol{P}_n \boldsymbol{k}_{i,n} = \boldsymbol{0}$. Consequently, for every $i \in \mathcal{P}$ we have $\delta \boldsymbol{u}_{\ell}(i) = \boldsymbol{0}$, and by Eqn. 7 the induced routing shift satisfies $\delta \boldsymbol{g}_{\ell}(i) \approx \boldsymbol{0}$, minimizing Eqn. 6 on the preservation set.

**Projected editing objective.** Let $\tilde{\boldsymbol{k}}_{i,n} = \boldsymbol{P}_n \boldsymbol{k}_{i,n}$ denote the projected key. Substituting $\Delta_n = \hat{\Delta}_n P_n$ into the MoE objective (Eqn. 5) yields

$$\{\hat{\boldsymbol{\Delta}}_n\}_{n=1}^N = \arg \min_{\{\tilde{\boldsymbol{\Delta}}_n\}} \sum_{i \in \mathcal{E}} \left\| \sum_{n=1}^N g_{i,n} \left( \boldsymbol{W}_n \boldsymbol{k}_{i,n} + \tilde{\boldsymbol{\Delta}}_n \tilde{\boldsymbol{k}}_{i,n} \right) - \boldsymbol{v}_i \right\|^2 + \lambda \sum_{n=1}^N \| \tilde{\boldsymbol{\Delta}}_n \|^2, \tag{8}$$

where $\lambda \geq 0$ controls the update magnitude and improves locality. No separate preservation term is needed, as $\boldsymbol{P}_n$ removes all preservation components by construction.

## 4.3 RANDOMIZED BLOCK COORDINATE DESCENT SOLVER

A naive step is to solve the projected objective in Eqn. 8 in one shot, just like what we do in dense model KE (Meng et al., 2022; 2023; Fang et al., 2025). In this subsection, we (i) expose the structure of the global closed-form solution, (ii) explain why the direct route is computationally impractical in MoE, and (iii) arrive at an efficient randomized block coordinate descent (BCD) procedure that scales with the expert hidden size.

**The global closed-form (one shot).** Let $\hat{\boldsymbol{\Delta}}_n \in \mathbb{R}^{d_m \times d_k}$ be the projected free variable for expert $n$, and stack all expert updates horizontally as $\hat{\boldsymbol{\Delta}} = [\hat{\boldsymbol{\Delta}}_1 \cdots \hat{\boldsymbol{\Delta}}_N] \in \mathbb{R}^{d_m \times (N d_k)}$. For edit example $i$, define the base residual (excluding any edits) $\boldsymbol{r}_i = \boldsymbol{v}_i - \sum_{n=1}^N g_{i,n} \boldsymbol{W}_n \boldsymbol{k}_{i,n}$, and the design vector

$\tilde{\psi}_i = [g_{i,1}\tilde{k}_{i,1}^\top \cdots g_{i,N}\tilde{k}_{i,N}^\top]^\top \in \mathbb{R}^{Nd_k}$, where $\tilde{k}_{i,n} = P_n k_{i,n}$. Then Eqn. 8 becomes a regularized multi-output linear regression: $\min_{\hat{\Delta}} \sum_{i\in\mathcal{E}} \|\hat{\Delta}\tilde{\psi}_i - r_i\|^2 + \lambda \sum_{n=1}^N \|\hat{\Delta}_n\|^2$. Vectorizing with $\theta = \text{vec}(\hat{\Delta}) \in \mathbb{R}^{d_m N d_k}$ and using $\text{vec}(\hat{\Delta}\tilde{\psi}_i) = (\tilde{\psi}_i^\top \otimes I_{d_m})\theta$, the normal equations take the compact form

$$\left(\sum_{i\in\mathcal{E}}(\tilde{\psi}_i\tilde{\psi}_i^\top) \otimes I_{d_m} + \lambda I_{d_m N d_k}\right)\theta = \sum_{i\in\mathcal{E}}(\tilde{\psi}_i \otimes I_{d_m})r_i, \tag{9}$$

with unique minimizer

$$\theta^\star = M_{\text{glob}}^{-1} b_{\text{glob}} \quad \text{and} \quad \hat{\Delta}^\star = \text{unvec}(\theta^\star), \tag{10}$$

where $M_{\text{glob}} = \sum_i(\tilde{\psi}_i\tilde{\psi}_i^\top) \otimes I_{d_m} + \lambda I$ and $b_{\text{glob}} = \sum_i(\tilde{\psi}_i \otimes I_{d_m})r_i$. A proof is provided in Appendix B.2.

Although Eqn. 9 is elegant, it is not a practical editing primitive at MoE scale. First, even exploiting the Kronecker structure, the system decomposes into $d_m$ independent problems of size $(Nd_k) \times (Nd_k)$ each. For typical MoE layers, $N$ can be 8–128 and $d_k$ in the thousands, so factorizing $d_m$ such systems (and re-factorizing as $\mathcal{E}$ changes) is prohibitively expensive in both time $O(d_m(Nd_k)^3)$ and memory $O(d_m(Nd_k)^2)$. Second, while Top-$K$ routing makes each $\tilde{\psi}_i$ $K$-block sparse, the accumulated Gram matrix $\sum_i \tilde{\psi}_i\tilde{\psi}_i^\top$ quickly densifies, yielding substantial fill-in under Cholesky/LDL$^\top$. These realities make the one-shot solve not suitable for fast, iterative MoE editing.

**From global to block: randomized BCD.** Since the insight that every expert is a natural chunk. The structure of Eqn. 8 suggests a block strategy: treat each expert as a block and optimize one block while holding the rest fixed. This reduces the problem to a sequence of well-conditioned, small ridge least-squares solves of size $d_k \times d_k$.

Fix $\{\hat{\Delta}_\ell\}_{\ell \neq n}$ and define the residual that excludes expert $n$:

$$r_i^{(-n)} = v_i - \sum_{\ell \neq n} g_{i,\ell}(W_\ell k_{i,\ell} + \hat{\Delta}_\ell \tilde{k}_{i,\ell}). \tag{11}$$

The subproblem in $\hat{\Delta}_n$ becomes the ridge-regularized least squares

$$\min_{\hat{\Delta}_n} \sum_{i\in\mathcal{E}} \|r_i^{(-n)} - g_{i,n}\hat{\Delta}_n\tilde{k}_{i,n}\|^2 + \lambda\|\hat{\Delta}_n\|^2. \tag{12}$$

Its normal equations

$$\hat{\Delta}_n \underbrace{\left(\sum_{i\in\mathcal{E}} g_{i,n}^2 \tilde{k}_{i,n}\tilde{k}_{i,n}^\top + \lambda I\right)}_{M_n \in \mathbb{R}^{d_k \times d_k}} = \underbrace{\left(\sum_{i\in\mathcal{E}} g_{i,n} r_i^{(-n)}\tilde{k}_{i,n}^\top\right)}_{B_n \in \mathbb{R}^{d_m \times d_k}}, \tag{13}$$

admit the closed-form update

$$\hat{\Delta}_n^\star = B_n M_n^{-1} = \left(\sum_{i\in\mathcal{E}} g_{i,n} r_i^{(-n)}\tilde{k}_{i,n}^\top\right)\left(\sum_{i\in\mathcal{E}} g_{i,n}^2 \tilde{k}_{i,n}\tilde{k}_{i,n}^\top + \lambda I\right)^{-1}. \tag{14}$$

We then write to parameters via the projection $\Delta_n^\star = \hat{\Delta}_n^\star P_n$ and move to the next block.

**Practicalities and complexity.** We traverse experts in randomized order and update only those active in the current minibatch, which further reduces cost. For each updated expert, forming $M_n$ costs $O(|\mathcal{E}|d_k^2)$ and inverting it costs $O(d_k^3)$, typically modest since $d_k \ll d_m$. We stream-accumulate $B_n$ and $M_n$, cache $\tilde{k}_{i,n}$, and use Cholesky with diagonal loading for numerical stability. Because Eqn. 8 is a strictly convex quadratic in $\{\hat{\Delta}_n\}$, (randomized) BCD with exact block solves converges globally under standard conditions (Tseng, 2001; Richtárik & Takáč, 2014). Empirically we see fast decrease within a few passes ($\leq 10$).

Table 1: Sequential knowledge editing on MoE LLMs. *Eff.*, *Gen.*, *Spe.* denote Efficacy, Generalization, Specificity; *Uti.* is their mean (higher is better ↑). Best in **bold**, second-best underlined.

| Method | Model | COUNTERFACT | | | | ZsRE | | | |
|---|---|---|---|---|---|---|---|---|---|
| | | Eff.↑ | Gen.↑ | Spe.↑ | Uti.↑ | Eff.↑ | Gen.↑ | Spe.↑ | Uti.↑ |
| Pre-edited | | $13.30_{\pm0.34}$ | $15.10_{\pm0.31}$ | $84.45_{\pm0.24}$ | 37.62 | $41.30_{\pm0.29}$ | $40.50_{\pm0.28}$ | $40.91_{\pm0.27}$ | 40.90 |
| FT | Qwen3-30B-A3B | $80.70_{\pm0.39}$ | $63.95_{\pm0.43}$ | $41.44_{\pm0.39}$ | 62.03 | $6.44_{\pm0.14}$ | $6.13_{\pm0.14}$ | $2.15_{\pm0.06}$ | 4.91 |
| FT-L | | $82.40_{\pm0.38}$ | $22.75_{\pm0.33}$ | $\underline{71.48}_{\pm0.25}$ | 58.88 | $\underline{44.19}_{\pm0.29}$ | $\underline{42.46}_{\pm0.29}$ | $\underline{41.92}_{\pm0.27}$ | $\underline{42.86}$ |
| AdaLoRA | | $51.90_{\pm0.50}$ | $49.75_{\pm0.40}$ | $48.10_{\pm0.26}$ | 49.92 | $3.66_{\pm0.09}$ | $3.60_{\pm0.09}$ | $4.68_{\pm0.10}$ | 3.98 |
| UnKE | | $\underline{89.30}_{\pm0.31}$ | $\underline{82.85}_{\pm0.33}$ | $48.15_{\pm0.33}$ | $\underline{73.43}$ | $31.43_{\pm0.28}$ | $29.78_{\pm0.27}$ | $25.30_{\pm0.23}$ | 28.84 |
| MoEEdit | | $\mathbf{99.30}_{\pm0.08}$ | $\mathbf{94.10}_{\pm0.20}$ | $\mathbf{80.97}_{\pm0.25}$ | $\mathbf{91.46}$ | $\mathbf{84.47}_{\pm0.22}$ | $\mathbf{78.01}_{\pm0.28}$ | $\mathbf{42.82}_{\pm0.28}$ | $\mathbf{68.43}$ |
| Pre-edited | | $11.80_{\pm0.32}$ | $14.70_{\pm0.31}$ | $84.53_{\pm0.24}$ | 37.01 | $33.20_{\pm0.28}$ | $32.14_{\pm0.28}$ | $28.02_{\pm0.00}$ | 31.12 |
| FT | GPT-OSS-20B | $\underline{83.40}_{\pm0.37}$ | $\mathbf{58.40}_{\pm0.42}$ | $55.72_{\pm0.33}$ | $\underline{65.84}$ | $25.57_{\pm0.28}$ | $23.41_{\pm0.26}$ | $17.61_{\pm0.21}$ | 22.20 |
| FT-L | | $73.80_{\pm0.44}$ | $43.10_{\pm0.48}$ | $59.75_{\pm0.33}$ | 58.88 | $32.75_{\pm0.29}$ | $33.09_{\pm0.30}$ | $30.06_{\pm0.26}$ | 31.97 |
| AdaLoRA | | $62.40_{\pm0.48}$ | $\underline{55.00}_{\pm0.42}$ | $43.65_{\pm0.34}$ | 53.68 | $43.46_{\pm0.30}$ | $42.96_{\pm0.30}$ | $\mathbf{33.60}_{\pm0.24}$ | 40.01 |
| UnKE | | $78.00_{\pm0.41}$ | $44.40_{\pm0.42}$ | $\underline{73.91}_{\pm0.28}$ | 65.44 | $\underline{46.58}_{\pm0.31}$ | $\underline{43.99}_{\pm0.31}$ | $31.40_{\pm0.26}$ | $\underline{40.66}$ |
| MoEEdit | | $\mathbf{95.90}_{\pm0.20}$ | $44.10_{\pm0.43}$ | $\mathbf{81.09}_{\pm0.25}$ | $\mathbf{73.70}$ | $\mathbf{81.68}_{\pm0.25}$ | $\mathbf{68.44}_{\pm0.34}$ | $\underline{32.55}_{\pm0.26}$ | $\mathbf{60.89}$ |

# 5 EXPERIMENTS

## 5.1 BASELINES, DATASETS, AND METRICS

We evaluate two modern MoE LLMs on standard factual-editing benchmarks: **Qwen3-30B-A3B** (Yang et al., 2025) (128 experts; top-8 per token) and **GPT-OSS-20B** (Agarwal et al., 2025) (32 experts; top-4 per token). As baselines, we adapt parameter-editing methods originally designed for dense Transformers but directly applicable to MoE models: Fine-Tuning (FT) (Zhu et al., 2020), FT-L (FT with a norm constraint), AdaLoRA (Zhang et al., 2023), and UnKE (Deng et al., 2025).

Following prior work, we use **COUNTERFACT** (single-hop counterfactual edits introduced with MEMIT) (Meng et al., 2023) and **ZsRE** (zero-shot relation extraction) (Levy et al., 2017). Visualized dataset examples are provided in Appendix D for unfamiliar readers. We report the standard editing metrics (Meng et al., 2022; 2023; Mitchell et al., 2022a): (i) **Efficacy** (edit success on edited prompts), (ii) **Generalization** (success on paraphrases and lightly perturbed contexts), and (iii) **Specificity** (locality on unrelated controls). Unless stated otherwise, we perform sequential batched edits. And to summarize the overall trade-off, we additionally report **Utility** as the mean of the three metrics. See Appendix A for full calculation details.

## 5.2 MAIN RESULTS ON KNOWLEDGE EDITING

We perform 1,000 sequential edits on each dataset (COUNTERFACT and ZsRE) with a batch size of 50 edits for all methods. Table 1 summarizes results on Qwen3-30B-A3B and GPT-OSS-20B. As shown in Table 1, MoEEdit consistently delivers outstanding results. On COUNTERFACT, it achieves over 90 efficacy on both backbones, substantially outperforming UnKE and FT-L in generalization and specificity, respectively. On GPT-OSS-20B, although FT attains slightly higher generalization, MoEEdit still provides the best overall balance, with clear gains in efficacy (+12.5) and specificity (+7.2). On ZsRE, MOEEDIT also demonstrates large improvements in efficacy and generalization, over +30 points against the strongest baselines, while maintaining competitive specificity (within 1 point of AdaLoRA). These results highlight that MOEEDIT offers the most favorable trade-off between accuracy and locality across models and datasets.

## 5.3 MAIN RESULTS ON ROUTING DISTRIBUTION SHIFT

We analyze routing distribution shifts under sequential editing. Similar to Section 5.2, we perform 1,000 edits with a batch size of 50. To control for depth, all methods are constrained to update at most the top editing layer (layer 7). Specifically, MoEEdit applies updates across layers {3,4,5,6,7} using BCD, while FT, FT-L, UnKE, and AdaLoRA are restricted to layer 7. We evaluate on Qwen3-30B-A3B / COUNTERFACT and report the routing-similarity (RS) metric between pre- and post-edit Top-$K$ expert sets (Eqn. 6), averaged over windows of 10 layers, for both the editing and preservation sets. The editing set consists of the 1,000 edited samples, while the preservation set is formed by

Table 2: Routing distribution shift on Qwen3-30B-A3B. Values are Jaccard similarity (**RS↑**) between pre- and post-edit routing distributions. Higher is better. Best results are in **bold**, second-best are underlined.

| Method | Model | Editing Set RS↑ | | | Preservation Set RS↑ | | |
|---|---|---|---|---|---|---|---|
| | | Lay. 11–20 | Lay. 21–30 | Lay. 31–40 | Lay. 11–20 | Lay. 21–30 | Lay. 31–40 |
| FT | | 23.57 | 26.58 | 29.98 | 24.72 | 27.45 | 30.97 |
| FT-L | | 47.01 | 51.20 | 53.68 | 48.80 | 50.17 | 53.45 |
| AdaLoRA | Qwen3-30B-A3B | 16.63 | 24.11 | 27.00 | 16.38 | 23.84 | 26.60 |
| UnKE | | 52.46 | 44.12 | 44.80 | 49.90 | 41.91 | 43.84 |
| MoEEdit | | **86.62** | **88.16** | **89.93** | **87.02** | **88.55** | **90.22** |

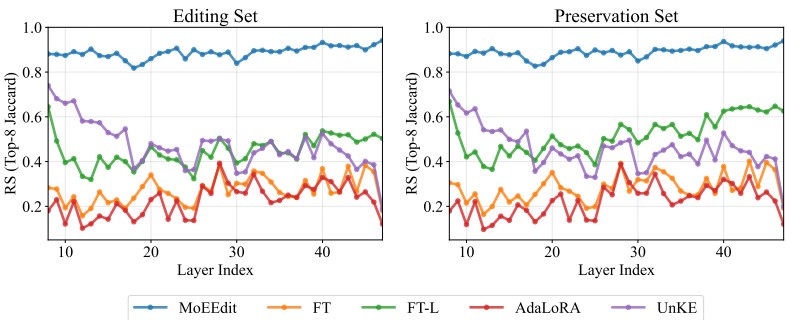

Figure 2: Routing similarity (RS) before and after editing on the editing and preservation sets. MoEEdit achieves consistently high RS, demonstrating strong routing stability.

sampling an equal number of untouched examples from the remaining dataset. Table 2 summarizes the results.

**Projection suppresses routing drift and preserves stability.** As shown in Table 2, methods directly extended from dense models exhibit substantial routing drift, whereas MoEEdit maintains consistently high routing stability (average RS > 88 across all layer ranges for both sets). Considering that Qwen3-30B-A3B activates 8 experts per token, the average number of non-overlapping experts before and after editing is close to one, which is negligible. This observation aligns with the heavy-tailed nature of routing: small perturbations primarily affect low-weight expert selections that contribute little to the output. The average KL divergence between pre- and post-edit routing distributions for MoEEdit is only 0.02, indicating minimal shift. Furthermore, Figure 2 plots routing similarity across layers for both editing and preservation sets. AdaLoRA and FT exhibit the lowest RS values across layers due to their unconstrained updates that heavily disrupt routing patterns. In contrast, MoEEdit preserves routing stability across all layers and consistently outperforms all baselines.

## 5.4  ABLATION STUDY

**Effect of Projection.** We ablate the projection matrix in MoEEdit to evaluate its contribution to routing stability. As shown in Table 3, removing projection reduces RS by an average of 14.81 points on the editing set and 15.21 on the preservation set. The KL divergence also increases from 0.02 to 0.0834, confirming that projection is critical for suppressing routing drift. These results validate the projection design introduced in Section 4.2.

Table 3: Ablation on the projection matrix. Removing projection significantly weakens routing stability.

| Method | Set | RS↑ | | |
|---|---|---|---|---|
| | | Lay. 11–20 | Lay. 21–30 | Lay. 31–40 |
| MoEEdit | Edit. | 86.62 | 88.16 | 89.93 |
| MoEEdit (w/o Proj) | | 73.64 | 72.90 | 73.75 |
| MoEEdit | Pres. | 87.02 | 88.55 | 90.22 |
| MoEEdit (w/o Proj) | | 73.59 | 73.08 | 73.50 |

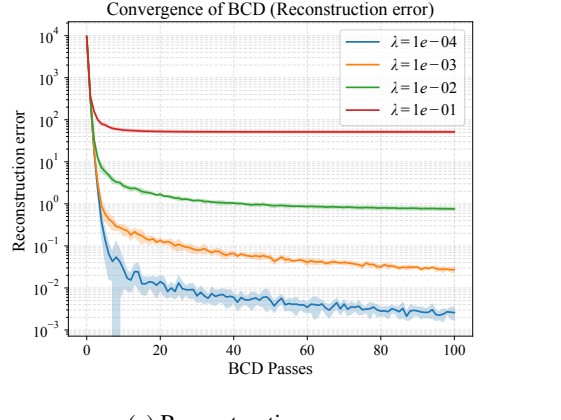 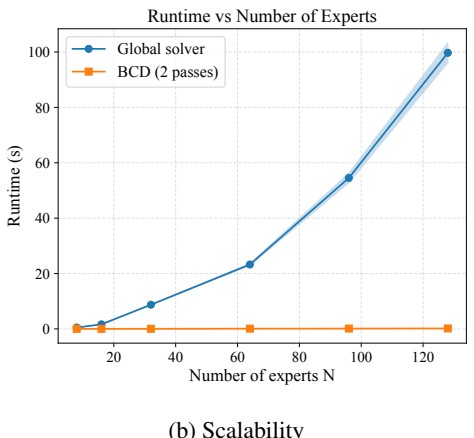

(a) Reconstruction error                    (b) Scalability

Figure 3: Comparison of solvers. (a) BCD achieves fast convergence with suitable $\lambda$. (b) BCD scales efficiently with the number of experts, while the closed-form solver quickly becomes infeasible.

**BCD Solver vs. Closed-form Solver.** We compare the proposed block coordinate descent (BCD) solver with a naive closed-form solution. The latter requires inverting large matrices that scale with the number of experts and constraints, making it computationally infeasible at realistic scales. To enable comparison, we construct a controlled synthetic batch and evaluate (i) convergence and (ii) scalability. Figure 3 illustrates two perspectives on our BCD solver: (a) reconstruction error convergence under varying $\lambda$, and (b) runtime scalability with respect to the number of experts. Smaller $\lambda$ values (e.g., $10^{-4}, 10^{-3}$) achieve lower error, whereas larger values converge to higher error floors. Panel (b) shows that the closed-form solver exhibits near-quadratic runtime growth and becomes infeasible beyond $N \approx 60$, while BCD maintains nearly constant runtime up to 128 experts. Thus, BCD scales linearly with hidden size rather than the total number of experts, ensuring practical efficiency.

**Number of BCD Passes.** We vary the number of BCD passes $\in \{2, 4, 6, 8, 10, 12, 14, 16, 18, 20\}$ while keeping all other settings fixed, and evaluate efficacy, generality, specificity, and their harmonic mean. As shown in Figure 4, as the number of BCD passes increases, both efficacy and generality rise rapidly in the early phase and then plateau, reflecting diminishing returns beyond moderate passes. Specificity shows a more stable, gradual upward trend. Since each subproblem is convex, early passes remove dominant residuals, while later passes only refine small residuals, yielding slower gains. This behavior suggests that a small number of passes (e.g., 6–10) already achieves a favorable trade-off between performance and efficiency. Beyond this range, additional passes

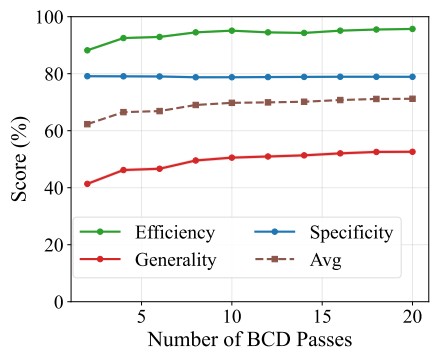

Figure 4: Ablation on the number of passes.

bring only marginal improvements while increasing runtime, highlighting the practicality of moderate BCD iterations in large-scale editing scenarios.

## 6 DISCUSSION AND CONCLUSION

In this work, we presented MoEEdit, a routing-stable knowledge editing framework tailored for Mixture-of-Experts (MoE) LLMs. Our approach addresses the unique challenges of computational cost, inter-expert coupling, and routing drift by combining per-expert null-space projection with an efficient block coordinate descent solver. Extensive experiments on COUNTERFACT and ZsRE benchmarks demonstrate that MoEEdit achieves high efficacy, strong generalization, and routing stability, all with superior efficiency compared to prior methods. Beyond empirical performance, our findings highlight several broader insights. First, expert-aware design is crucial: naive adaptations

of dense-model editors to MoEs fail to maintain stability and efficiency. Second, routing stability emerges as a central factor in editing sparse architectures, where even small perturbations can cascade through routing distributions. By explicitly controlling for this effect, MoEEdit offers a principled solution that preserves locality without sacrificing scalability. In summary, MoEEdit establishes a robust foundation for precise and scalable knowledge editing in sparse architectures, advancing the state of the art and paving the way for more adaptive and trustworthy MoE-based language models.

## ACKNOWLEDGEMENT

R. Wei and P. Li are partially supported by the NSF under awards PHY-2117997, IIS-2239565, IIS-2428777, and CCF-2402816; the JPMorgan Chase Faculty Award; and the OpenAI Researcher Access Program Credit, the Nvidia Academic Award, and the Google Academic Award.

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

## A    METRICS

In this section, we introduce the evaluation metrics we use for COUNTERFACT and ZsRE

### A.1    ZsRE EVALUATION METRICS

Building on prior studies, we define for each ZsRE metric a large language model $\mathcal{M}$, a factual prompt $(s_i, r_i)$, a revised target output $y_i$, and the model's initial output $\hat{y}_i$:

**Effectiveness.** Effectiveness is quantified as the average top-1 accuracy on edited inputs:

$$\mathbb{E}_i\Big[\mathbf{1}\Big(y_i = \arg\max_y \Pr_{\mathcal{M}}(y \mid (s_i, r_i))\Big)\Big]. \tag{15}$$

**Generalization.** This measures the ability of the model to perform correctly on paraphrased prompts $\tilde{N}(s_i, r_i)$. It is computed as the average accuracy over such rephrasings:

$$\mathbb{E}_i\Big[\mathbf{1}\Big(y_i = \arg\max_y \Pr_{\mathcal{M}}(y \mid \tilde{N}(s_i, r_i))\Big)\Big]. \tag{16}$$

**Specificity.** Specificity ensures that modifications do not affect unrelated cases $\Omega(s_i, r_i)$. It is defined as:

$$\mathbb{E}_i\Big[\mathbf{1}\Big(\hat{y}_i = \arg\max_y \Pr_{\mathcal{M}}(y \mid \Omega(s_i, r_i))\Big)\Big]. \tag{17}$$

### A.2    COUNTERFACTUAL EVALUATION METRICS

Similarly, we introduce Counterfactual metrics for $\mathcal{M}$ under the same setup $(s_i, r_i)$ with target $y_i$ and original $\hat{y}_i$:

**Effectiveness (success ratio).** The share of cases where $y_i$ is assigned higher probability than $\hat{y}_i$ under $(s_i, r_i)$:

$$\mathbb{E}_i\Big[\Pr_{\mathcal{M}}(y_i \mid (s_i, r_i)) > \Pr_{\mathcal{M}}(\hat{y}_i \mid (s_i, r_i))\Big]. \tag{18}$$

**Generalization (paraphrase success).** The proportion of paraphrased prompts $\tilde{N}(s_i, r_i)$ where $y_i$ has higher likelihood than $\hat{y}_i$:

$$\mathbb{E}_i\Big[\Pr_{\mathcal{M}}(y_i \mid \tilde{N}(s_i, r_i)) > \Pr_{\mathcal{M}}(\hat{y}_i \mid \tilde{N}(s_i, r_i))\Big]. \tag{19}$$

**Specificity (neighborhood success).** For neighborhood prompts $\Omega(s_i, r_i)$ that involve related but distinct entities, specificity is the fraction of cases where $y_i$ is favored over $\hat{y}_i$:

$$\mathbb{E}_i\Big[\Pr_{\mathcal{M}}(y_i \mid \Omega(s_i, r_i)) > \Pr_{\mathcal{M}}(\hat{y}_i \mid \Omega(s_i, r_i))\Big]. \tag{20}$$

## B    PROOF & KNOWLEDGE EDITING

### B.1    GENERAL FRAMEWORK OF LOCATE-THEN-EDIT

Knowledge editing seeks to precisely revise a model's behavior to recall a new fact $(s, r, o^*)$ in place of an obsolete or incorrect one $(s, r, o)$, conditioned on a prompt $p(s, r)$. While various techniques exist, the dominant *locate-then-edit* paradigm (Meng et al., 2022; 2023) typically decomposes the process into three distinct phases: locating the mediating parameters, computing the optimal local update targets, and solving for the new weights. We formalize this process below using notation consistent with Section 3.

**Step 1: Causal Localization.** The initial phase identifies the specific layer $l$ and module (e.g., a dense FFN or specific experts in an MoE layer) that mediate the retrieval of the factual association. This is commonly achieved via *Causal Tracing* (Meng et al., 2022). By corrupting hidden states

with noise to degrade the model's prediction and subsequently restoring states at specific layers, one can quantify the *Indirect Effect* (IE) of each layer on the correct output probability. The layer $l^*$ exhibiting the maximal causal influence is selected as the target for editing:

$$l^* = \arg\max_l \text{IE}(l). \tag{21}$$

**Step 2: Acquiring the Target Output Vector ($v^*$).** Once the target layer $l^*$ is identified, we must determine the optimal output representation required to successfully trigger the target token $o^*$. Viewing the layer as a linear associative memory, it maps a key $k$ (input features) to a value $v$ (output features). The objective is to identify a new output vector $v^*$ such that, if the layer were to produce $v^*$, the final model prediction would be $o^*$.

This step is formulated as an optimization problem over the hidden state vector rather than the model parameters. Let $G(v)$ denote the function mapping the layer output $v$ to the final model logits. We freeze the model parameters and optimize a perturbation $\delta$ to the original output $v$, aiming to maximize the log-likelihood of the target object $o^*$:

$$v^* = v + \delta^*, \quad \text{where} \quad \delta^* = \arg\min_\delta -\log \mathbb{P}_\mathcal{M} \left( o^* \mid \text{do}(v \leftarrow v + \delta) \right). \tag{22}$$

Here, the $\text{do}(\cdot)$ operator signifies a causal intervention where the layer output is manually set to $v + \delta$. A regularization term (e.g., KL divergence) is typically included in the objective to minimize prediction drift for the subject $s$ and relation $r$, ensuring the edit remains semantically consistent. This process effectively translates the semantic edit target (the token $o^*$) into a vector-space target $v^*$.

**Step 3: Updating Parameters.** The final step is to update the projection weights $W$ (corresponding to $W_{out}$ in dense models or $\{W_n\}$ in MoE experts) to map the specific input key $k$ to the new target $v^*$, while preserving unrelated associations. This is formulated as a constrained least-squares problem.

Let $\mathcal{E}$ denote the set of edit examples (new facts) and $\mathcal{P}$ denote the set of preservation examples (invariant knowledge). We require $Wk_i \approx v_i^*$ for edits $i \in \mathcal{E}$, and $Wk_j \approx v_j$ for preservation samples $j \in \mathcal{P}$. The optimal update $\hat{W}$ minimizes the aggregated error:

$$\hat{W} = \arg\min_W \sum_{i \in \mathcal{E}} \|Wk_i - v_i^*\|^2 + \sum_{j \in \mathcal{P}} \|Wk_j - v_j\|^2. \tag{23}$$

Dense methods such as ROME and MEMIT solve this globally via a closed-form solution involving the covariance matrix of the keys. As discussed in Section 4, our proposed MoEEdit adapts this general objective to address the unique constraints of Mixture-of-Experts architectures.

## B.2 SUPPLEMENTARY PROOF

### B.2.1 PROOF OF THE GLOBAL CLOSED-FORM (ONE-SHOT) SOLUTION

Let $\{\tilde{\boldsymbol{k}}_{i,n}\}_{i \in \mathcal{E}, n \in [N]}$ be the projected keys and $g_{i,n} \geq 0$ the router weights. For each edit example $i \in \mathcal{E}$, define the base residual

$$\boldsymbol{r}_i = \boldsymbol{v}_i - \sum_{n=1}^{N} g_{i,n} \boldsymbol{W}_n \boldsymbol{k}_{i,n}, \tag{24}$$

and the design vector

$$\tilde{\boldsymbol{\psi}}_i = \left[ g_{i,1} \tilde{\boldsymbol{k}}_{i,1}^\top \quad \cdots \quad g_{i,N} \tilde{\boldsymbol{k}}_{i,N}^\top \right]^\top \in \mathbb{R}^{Nd_k}. \tag{25}$$

Stack the expert updates as $\hat{\boldsymbol{\Delta}} = [\hat{\boldsymbol{\Delta}}_1 \cdots \hat{\boldsymbol{\Delta}}_N] \in \mathbb{R}^{d_m \times (Nd_k)}$. The projected objective

$$\min_{\hat{\boldsymbol{\Delta}}} \sum_{i \in \mathcal{E}} \left\| \hat{\boldsymbol{\Delta}} \tilde{\boldsymbol{\psi}}_i - \boldsymbol{r}_i \right\|_2^2 + \lambda \sum_{n=1}^{N} \left\| \hat{\boldsymbol{\Delta}}_n \right\|_F^2 \tag{26}$$

has the unique minimizer

$$\theta^\star = \boldsymbol{M}_{\text{glob}}^{-1} \boldsymbol{b}_{\text{glob}}, \qquad \hat{\boldsymbol{\Delta}}^\star = \text{unvec}(\theta^\star), \tag{27}$$

with

$$\boldsymbol{M}_{\text{glob}} = \left(\sum_{i\in\mathcal{E}} \tilde{\boldsymbol{\psi}}_i \tilde{\boldsymbol{\psi}}_i^\top\right) \otimes \boldsymbol{I}_{d_m} + \lambda \boldsymbol{I}_{d_m N d_k}, \qquad \boldsymbol{b}_{\text{glob}} = \sum_{i\in\mathcal{E}} \left(\tilde{\boldsymbol{\psi}}_i \otimes \boldsymbol{I}_{d_m}\right) \boldsymbol{r}_i. \tag{28}$$

Let $m := |\mathcal{E}|$ and define the data matrices

$$\boldsymbol{\Psi} = \begin{bmatrix} \tilde{\boldsymbol{\psi}}_1 \cdots \tilde{\boldsymbol{\psi}}_m \end{bmatrix} \in \mathbb{R}^{N d_k \times m}, \quad \boldsymbol{R} = \begin{bmatrix} \boldsymbol{r}_1 \cdots \boldsymbol{r}_m \end{bmatrix} \in \mathbb{R}^{d_m \times m}. \tag{29}$$

The objective becomes

$$\min_{\hat{\boldsymbol{\Delta}} \in \mathbb{R}^{d_m \times (N d_k)}} \left\| \hat{\boldsymbol{\Delta}} \boldsymbol{\Psi} - \boldsymbol{R} \right\|_F^2 + \lambda \left\| \hat{\boldsymbol{\Delta}} \right\|_F^2. \tag{30}$$

Taking derivatives gives

$$\nabla_{\hat{\boldsymbol{\Delta}}} = 2(\hat{\boldsymbol{\Delta}} \boldsymbol{\Psi} - \boldsymbol{R}) \boldsymbol{\Psi}^\top + 2\lambda \hat{\boldsymbol{\Delta}}. \tag{31}$$

Setting this to zero yields

$$\hat{\boldsymbol{\Delta}} \left( \boldsymbol{\Psi} \boldsymbol{\Psi}^\top + \lambda \boldsymbol{I}_{N d_k} \right) = \boldsymbol{R} \boldsymbol{\Psi}^\top. \tag{32}$$

Using $\text{vec}(\boldsymbol{X}\boldsymbol{A}) = (\boldsymbol{A}^\top \otimes \boldsymbol{I})\,\text{vec}(\boldsymbol{X})$, Eqn. 32 becomes

$$\left( (\boldsymbol{\Psi}\boldsymbol{\Psi}^\top) \otimes \boldsymbol{I}_{d_m} + \lambda \boldsymbol{I}_{d_m N d_k} \right) \boldsymbol{\theta} = \text{vec}(\boldsymbol{R}\boldsymbol{\Psi}^\top), \tag{33}$$

where $\boldsymbol{\theta} = \text{vec}(\hat{\boldsymbol{\Delta}})$. Noting that $\text{vec}(\boldsymbol{R}\boldsymbol{\Psi}^\top) = \sum_{i=1}^m (\tilde{\boldsymbol{\psi}}_i \otimes \boldsymbol{I}_{d_m})\boldsymbol{r}_i$, we obtain the system in the theorem statement.

$\boldsymbol{M}_{\text{glob}} = (\boldsymbol{\Psi}\boldsymbol{\Psi}^\top) \otimes \boldsymbol{I}_{d_m} + \lambda\boldsymbol{I}$ is positive definite for $\lambda > 0$, hence invertible. The minimizer is unique.

Thus $\boldsymbol{\theta}^\star = \boldsymbol{M}_{\text{glob}}^{-1}\boldsymbol{b}_{\text{glob}}$, and reshaping yields $\hat{\boldsymbol{\Delta}}^\star = \text{unvec}(\boldsymbol{\theta}^\star)$.

### B.2.2 DETAILED DERIVATION OF THE SINGLE-EXPERT SUBPROBLEM

Starting from the projected MoE objective (Eqn. 8),

$$\min_{\{\hat{\Delta}_n\}_{n=1}^N} \sum_{i\in\mathcal{E}} \left\| \sum_{n=1}^N g_{i,n}\big(\boldsymbol{W}_n \boldsymbol{k}_{i,n} + \hat{\Delta}_n \tilde{\boldsymbol{k}}_{i,n}\big) - \boldsymbol{v}_i \right\|^2 + \lambda \sum_{n=1}^N \|\hat{\Delta}_n\|^2, \tag{34}$$

we apply block coordinate descent (BCD) over experts, updating one expert at a time and keeping the others fixed. For a fixed expert $n$, collect all terms that do *not* involve $\hat{\Delta}_n$ into the external residual

$$\boldsymbol{r}_i^{(-n)} = \boldsymbol{v}_i - \sum_{\ell \neq n} g_{i,\ell}\big(\boldsymbol{W}_\ell \boldsymbol{k}_{i,\ell} + \hat{\Delta}_\ell \tilde{\boldsymbol{k}}_{i,\ell}\big), \tag{35}$$

and substitute Eqn. 35 back into Eqn. 34. The single-expert subproblem for $\hat{\Delta}_n$ is the ridge-regularized least-squares

$$\min_{\hat{\Delta}_n} \sum_{i\in\mathcal{E}} \left\| \boldsymbol{r}_i^{(-n)} - g_{i,n}\hat{\Delta}_n \tilde{\boldsymbol{k}}_{i,n} \right\|^2 + \lambda \|\hat{\Delta}_n\|^2. \tag{36}$$

Introduce compact notation

$$X = \hat{\Delta}_n, \quad x_i = \tilde{\boldsymbol{k}}_{i,n}, \quad y_i = \boldsymbol{r}_i^{(-n)}, \quad g_i = g_{i,n}, \tag{37}$$

so that

$$f(X) = \sum_{i\in\mathcal{E}} \left\| y_i - g_i X x_i \right\|^2 + \lambda \|X\|^2. \tag{38}$$

Using the standard matrix derivative identity[3] yields

$$\nabla f(X) = -2\sum_i g_i y_i x_i^\top + 2\sum_i g_i^2 X x_i x_i^\top + 2\lambda X. \tag{39}$$

---

[3]For vectors $a, b$ and matrix $X$, $\partial \|a - Xb\|^2 / \partial X = -2(a - Xb)b^\top$.

Setting the gradient to zero gives the normal equations

$$X\left(\sum_i g_i^2 x_i x_i^\top + \lambda I\right) = \sum_i g_i y_i x_i^\top. \tag{40}$$

Restoring expert-specific symbols, define

$$\boldsymbol{M}_n \triangleq \sum_{i\in\mathcal{E}} g_{i,n}^2 \tilde{\boldsymbol{k}}_{i,n} \tilde{\boldsymbol{k}}_{i,n}^\top + \lambda I, \qquad \boldsymbol{B}_n \triangleq \sum_{i\in\mathcal{E}} g_{i,n} \boldsymbol{r}_i^{(-n)} \tilde{\boldsymbol{k}}_{i,n}^\top. \tag{41}$$

Then Eqn. 40 becomes $\hat{\Delta}_n \boldsymbol{M}_n = \boldsymbol{B}_n$, with the unique minimizer

$$\hat{\Delta}_n^\star = \boldsymbol{B}_n \boldsymbol{M}_n^{-1}. \tag{42}$$

Since the actual expert update is parameterized via the projection $\Delta_n = \hat{\boldsymbol{\Delta}}_n \mathbf{P}_n$ (Sec. 4.2), the written update is

$$\boldsymbol{\Delta}_n^\star = \hat{\boldsymbol{\Delta}}_n^\star \mathbf{P}_n. \tag{43}$$

**Why $M_n$ is invertible (positive definite).** By definition,

$$\boldsymbol{M}_n = \underbrace{\sum_{i\in\mathcal{E}} g_{i,n}^2 \tilde{\boldsymbol{k}}_{i,n} \tilde{\boldsymbol{k}}_{i,n}^\top}_{\text{Gram matrix } \boldsymbol{G}_n \succeq 0} + \lambda I_{d_k}. \tag{44}$$

For any nonzero $z \in \mathbb{R}^{d_k}$,

$$z^\top \boldsymbol{M}_n z = \sum_{i\in\mathcal{E}} g_{i,n}^2 \left(z^\top \tilde{\boldsymbol{k}}_{i,n}\right)^2 + \lambda \|z\|^2. \tag{45}$$

Since $\lambda > 0$, then $z^\top \boldsymbol{M}_n z \geq \lambda \|z\|^2 > 0$ for all $z \neq 0$, so $\boldsymbol{M}_n \succ 0$ and is invertible. Moreover, $\lambda_{\min}(\boldsymbol{M}_n) \geq \lambda$, which ensures good conditioning (Tikhonov regularization). Because the projected keys are $\tilde{\boldsymbol{k}}_{i,n} = \mathbf{P}_n \boldsymbol{k}_{i,n}$ with an idempotent projector $\mathbf{P}_n$, they lie in $\text{range}(\mathbf{P}_n)$. When $\lambda > 0$, $\boldsymbol{M}_n$ remains strictly positive definite *on the full ambient space* (not only on $\text{range}(\mathbf{P}_n)$), guaranteeing a unique closed-form update 42.

## C EXPERIMENTS SETUP

**Model Configuration** We evaluate our method on two Mixture-of-Experts (MoE) models: Qwen3-30B-A3B[4] and GPT-OSS-20B[5]. Qwen3-30B-A3B contains 128 experts per layer with the top-8 experts activated per token, resulting in approximately 3.3B active parameters during inference. The latter, GPT-OSS-20B, features 32 experts per layer with the top-4 experts activated, corresponding to approximately 3.6B active parameters.

**Hardware and Quantization** All experiments are conducted on a single node equipped with an NVIDIA H20 GPU. To balance precision and memory constraints, we utilize the BF16 format for model weights, while optimization is performed in FP32 to ensure numerical stability.

**Fine-Tuning (FT)** We evaluate both standard Fine-Tuning (FT) and Constrained Fine-Tuning (FT-L). The primary distinction is that FT-L imposes a norm constraint $\varepsilon$ on the weight update. For both Qwen3-30B-A3B and GPT-OSS-20B, we set $\varepsilon = 1 \times 10^{-3}$ for FT-L. We adopt a learning rate of $1 \times 10^{-3}$ for both models. Updates are applied to layer 30 for Qwen3-30B-A3B and layer 0 for GPT-OSS-20B. For both methods, we target the mlp.experts.down_proj module. We train for 25 epochs, setting both weight decay and the KL divergence factor to 0.

**UnKE** As UnKE employs a two-stage structuring process, we configure the models as follows: For Qwen3-30B-A3B, the first stage uses a learning rate of $5 \times 10^{-1}$ with 25 optimization steps and a weight decay coefficient of $1 \times 10^{-3}$. In the second stage, we apply a learning rate of $2 \times 10^{-4}$ and perform 50 optimization steps. For GPT-OSS-20B, the first stage similarly adopts a learning rate of $5 \times 10^{-1}$ but with 50 optimization steps, utilizing the same weight decay ($1 \times 10^{-3}$). The second

---

[4]https://huggingface.co/Qwen/Qwen3-30B-A3B
[5]https://huggingface.co/openai/gpt-oss-20b

stage proceeds with a learning rate of $1 \times 10^{-4}$ and 50 optimization steps. All experiments restrict parameter updates to layer 7. Consistent with the focus on structured knowledge editing, optimization is performed on the last subject token for both models.

**AdaLoRA** For AdaLoRA, updates are applied across all layers. We set the hyperparameters $\alpha = 32$ and rank $r = 8$. For Qwen3-30B-A3B, we set the learning rate to $5 \times 10^{-3}$, while for GPT-OSS-20B, we use a reduced learning rate of $5 \times 10^{-4}$. The optimization is run for 25 steps for both models.

**MoEEdit (Ours)** For Qwen3-30B-A3B, we edit layers $\{3, 4, 5, 6, 7\}$, whereas for GPT-OSS-20B, we target layer 5. For Qwen3-30B-A3B, we perform 25 optimization steps with a learning rate of 0.1 and execute 4 Block Coordinate Descent (BCD) passes. For GPT-OSS-20B, we perform 50 optimization steps with a learning rate of 0.2 and 10 BCD passes. For both models, we set the regularization parameter $\lambda = 1$ and the KL factor to $0.0625$. We utilize 100,000 samples to compute the covariance matrix for the null-space projection with projection threshold = 0.02.

## D EXAMPLES OF ZSRE AND COUNTERFACT

```
{
  "subject": "Watts Humphrey",
  "src": "What university did Watts Humphrey attend?",
  "pred": "Trinity College",
  "rephrase": "What university did Watts Humphrey take part in?",
  "alt": "University of Michigan",
  "answers": [
    "Illinois Institute of Technology"
  ],
  "loc": "nq question: who played desmond doss father in hacksaw ridge",
  "loc_ans": "Hugo Weaving",
  "cond": "Trinity College >> University of Michigan || What university did
      Watts Humphrey attend?"
},
{
  "subject": "Ramalinaceae",
  "src": "Which family does Ramalinaceae belong to?",
  "pred": "Ramalinales",
  "rephrase": "What family are Ramalinaceae?",
  "alt": "Lamiinae",
  "answers": [
    "Lecanorales"
  ],
  "loc": "nq question: types of skiing in the winter olympics 2018",
  "loc_ans": "Downhill",
  "cond": "Ramalinales >> Lamiinae || Which family does Ramalinaceae belong
      to?"
},
{
  "subject": "Denny Herzig",
  "src": "What role does Denny Herzig play in football?",
  "pred": "midfielder",
  "rephrase": "What's Denny Herzig's role in football?",
  "alt": "winger",
  "answers": [
    "defender"
  ],
  "loc": "nq question: where does aarp fall on the political spectrum",
  "loc_ans": "non-partisan",
  "cond": "midfielder >> winger || What role does Denny Herzig play in
      football?"
}
```

Figure 5: Examples of ZsRE dataset

```
{
    "case_id": 975,
    "pararel_idx": 17275,
    "requested_rewrite": {
      "prompt": "{}, from",
      "relation_id": "P127",
      "target_new": {
        "str": "Google",
        "id": "Q95"
      },
      "target_true": {
        "str": "Microsoft",
        "id": "Q2283"
      },
      "subject": "Bing Videos"
    },
    "paraphrase_prompts": [
      "\"Old Jennifer: I'm $adjectiveOld!\" Bing Videos is owned by",
      "J. Bing Videos is from"
    ],
    "neighborhood_prompts": [
      "OneDrive is from",
      "German Research Center for Artificial Intelligence's owner",
      "Groove Music's owner",
      "Arkane Studios, from",
      "Yammer is from",
      "Yammer, by",
      "Turn 10 Studios, by",
      "German Research Center for Artificial Intelligence is owned by",
      "Mojang Studios is from",
      "id Software's owner"
    ]
}
```

Figure 6: An example of COUNTERFACT

