# OpenReview forum: "MoEEdit: Efficient and Routing-Stable Knowledge Editing for Mixture-of-Experts LLMs"
_ICLR.cc/2026/Conference — ICLR 2026 Poster_

### Official Review · Reviewer_fg5e · 2025-10-27

**Soundness:** 3
**Presentation:** 3
**Contribution:** 3
**Rating:** 6
**Confidence:** 3

**Summary:**

This paper proposes MoEEdit, a novel framework for knowledge editing in sparse Mixture-of-Experts (MoE) large language models. To address the challenges of high computational cost, expert coupling, and routing distribution shift when applying existing methods designed for dense models, MoEEdit introduces two key techniques: 1) per-expert null-space projection, which preserves outputs on retained samples and prevents routing shifts; and 2) a randomized block coordinate descent (BCD) solver that decomposes the large-scale optimization into smaller, expert-wise subproblems, improving computational efficiency. This work establishes a strong baseline for knowledge editing in MoE-based models.

**Strengths:**

Strength 1: The identification and analysis of "routing distribution shift" is a key contribution of this paper. The authors clearly explain how parameter edits in one MoE layer can trigger cascading changes in downstream routing patterns, thereby disrupting the model's expert specialization pathways, providing a theoretical foundation for future research.

Strength 2: The experiments are relatively comprehensive and demonstrate superior performance, effectively supporting the theoretical analysis.

**Weaknesses:**

Weakness 1: The actual computational cost of the BCD solver is not clearly presented, especially in comparison with traditional approaches—for example, the total time required to complete 1,000 edits. This lack of comparison could undermine the claim of "efficiency" made in the title.

Weakness 2: Experiments are conducted with up to 1,000 sequential edits. While this is a reasonable benchmark, the true stress test for knowledge editing lies in much larger scales (e.g., 3k or more) of continuous edits. It remains unclear whether MoEEdit’s performance and routing stability would degrade after such extensive editing.

**Questions:**

Question1: see Weakness 1
Question2: see Weakness 2
Question3: Have the authors considered or analyzed how the sampling strategy and size of the preservation set affect the model's editing performance?

---

> ### Author Response · Authors · 2025-11-22
> **Response to Reviewer fg5e**
>
> We thank the reviewer for their constructive feedback and for recognizing the novelty of our framework, particularly the identification of "routing distribution shift" and the theoretical foundation provided by our analysis. We appreciate the opportunity to clarify the efficiency of our solver and the scalability of our method.
>
> Below, we address your specific questions regarding computational cost, large-scale editing, and preservation set sensitivity.
>
> ---
>
> > **W1&Q1: The actual computational cost of the BCD solver is not clearly presented, especially in comparison with traditional approaches-for example, the total time required to complete 1,000 edits. This lack of comparison could undermine the claim of "efficiency" made in the title.**
>
> We appreciate the opportunity to clarify our efficiency claims. Our definition of efficiency is established in comparison to the global closed-form solution, which represents the general standard in dense model editing (e.g., ROME, MEMIT) but becomes intractable when applied to MoEs.
>
> In the locate-then-edit paradigm for dense models, the objective is typically solved globally to achieve optimality. However, as detailed in Section 4.3, naively applying this global solving paradigm to an MoE layer necessitates inverting a massive matrix of size $(N \cdot d_k) \times (N \cdot d_k)$. For a model with $N=128$ experts, this is computationally prohibitive and causes memory exhaustion, as demonstrated in our scalability analysis (Figure 3b). Our BCD solver decomposes this intractable global problem into manageable sub-problems, reducing complexity to scale linearly with the hidden size ($d_k$) rather than the total expert count.
>
> To demonstrate the practical runtime of this efficient solver, we measured the average wall-clock time for the BCD optimization process on a single batch (50 edits) when performing 1,000 sequential edits. The "BCD Opt. Time" refers to the time required to solve the linear system for the batch.
>
> **Table R1: Runtime of BCD Optimization (Per Batch of 50 Edits)**
>
> | **Dataset** | **Model** | **BCD Opt. Time** | **Time Per BCD Pass** |
> | --------------- | ------------- | ----------------- | --------------------- |
> | **ZsRE** | Qwen3-30B-A3B | 31.9s             | 1.6s                  |
> | **ZsRE** | GPT-OSS-20B   | 8.0s              | 0.8s                  |
> | **COUNTERFACT** | Qwen3-30B-A3B | 31.8s             | 1.6s                  |
> | **COUNTERFACT** | GPT-OSS-20B   | 7.9s              | 0.8s                  |
>
> The results demonstrate the computational efficiency of our proposed solver. For the large-scale Qwen3-30B model, the optimization for a batch of 50 edits is completed in approximately 32 seconds. With a latency of <2 seconds per optimization pass, the BCD solver scales effectively and facilitates practical deployment.

---

> ### Author Response · Authors · 2025-11-22
> **Response to Reviewer fg5e**
>
> > **W2&Q2: Experiments are conducted with up to 1,000 sequential edits. While this is a reasonable benchmark, the true stress test for knowledge editing lies in much larger scales (e.g., 3k or more) of continuous edits. It remains unclear whether MoEEdit’s performance and routing stability would degrade after such extensive editing.**
>
> This is an excellent suggestion. To test the limits of MoEEdit, we extended our sequential editing experiment to 2,000 and 5,000 edits on the Qwen3-30B-A3B model (COUNTERFACT dataset), maintaining a batch size of 50.
>
> **Table R2: Performance under Extended Sequential Editing (Qwen3-30B-A3B)**
>
> | **Edit Number** | **Eff.**   | **Loc.**   | **Spec.**  | **Uti.** |
> | --------------- | ---------- | ---------- | ---------- | -------- |
> | **1,000**       | 99.30±0.08 | 94.10±0.20 | 80.97±0.25 | 91.46    |
> | **2,000**       | 99.05±0.10 | 87.75±0.28 | 79.64±0.26 | 88.81    |
> | **5,000**       | 96.34±0.19 | 78.53±0.36 | 76.93±0.27 | 83.93    |
>
> **Table R3: Routing Stability (RS) under Extended Sequential Editing**
>
> | **Edit Number** | **Editing Set RS** |            |            | **Preservation Set RS** |            |            |
> | --------------- | ------------------ | ---------- | ---------- | ----------------------- | ---------- | ---------- |
> |                 | Lay. 11-20         | Lay. 21-30 | Lay. 31-40 | Lay. 11-20              | Lay. 21-30 | Lay. 31-40 |
> | **1,000**       | 86.62              | 88.16      | 89.93      | 87.02                   | 88.55      | 90.22      |
> | **2,000**       | 84.96              | 87.08      | 89.57      | 85.65                   | 87.31      | 90.06      |
> | **5,000**       | 81.49              | 84.42      | 86.84      | 81.35                   | 84.28      | 86.60      |
>
> The tables show that
>
> 1. Even after 5,000 edits, the efficacy remains remarkably high (>96%), indicating that the model retains the capacity to accept new knowledge.
> 2. Crucially, the Routing Stability (RS) degrades very gracefully. After 5,000 edits, the RS remains above 80% across all layers. This confirms that our null-space projection mechanism successfully prevents the "cascading routing shift" even under extreme editing loads.
> 3. We observe a moderate drop in Locality and Specificity (~4 points in Utility from 1k to 2k), which is a known phenomenon in continual learning/editing (saturation of the weight space). However, the model avoids catastrophic collapse.

---

> ### Author Response · Authors · 2025-11-22
> **Response to Reviewer fg5e**
>
> > **Q3: Have the authors considered or analyzed how the sampling strategy and size of the preservation set affect the model’s editing performance?**
>
> We employ a random sampling strategy from a general corpus (Wikipedia) to construct the preservation set $K_{n}^0$. To evaluate sensitivity to the set size, we performed an ablation study reducing the sample size from our default 100,000 down to 10,000.
>
> **Table R4: Effect of Preservation Set Size on Editing Performance**
>
> | **Samples (N)** | **COUNTERFACT** |          |           |          | **ZsRE** |          |           |          |
> | --------------- | --------------- | -------- | --------- | -------- | -------- | -------- | --------- | -------- |
> |                 | **Eff.**        | **Loc.** | **Spec.** | **Uti.** | **Eff.** | **Loc.** | **Spec.** | **Uti.** |
> | **10,000**      | 99.50           | 92.90    | 80.60     | 91.00    | 84.34    | 76.95    | 42.37     | 67.89    |
> | **20,000**      | 99.40           | 93.90    | 80.99     | 91.43    | 84.56    | 77.45    | 42.27     | 68.09    |
> | **50,000**      | 99.50           | 93.75    | 80.96     | 91.40    | 84.16    | 77.46    | 42.66     | 68.09    |
> | **100,000**     | 99.30           | 94.10    | 80.97     | 91.46    | 84.47    | 78.01    | 42.82     | 68.43    |
>
> **Table R5: Effect of Preservation Set Size on Routing Stability**
>
> | **Samples (N)** | **Editing Set RS** |            |            | **Preservation Set RS** |            |            |
> | --------------- | ------------------ | ---------- | ---------- | ----------------------- | ---------- | ---------- |
> |                 | Lay. 11-20         | Lay. 21-30 | Lay. 31-40 | Lay. 11-20              | Lay. 21-30 | Lay. 31-40 |
> | **10,000**      | 87.04              | 88.42      | 89.38      | 87.41                   | 88.60      | 89.53      |
> | **20,000**      | 86.25              | 88.26      | 89.96      | 86.63                   | 88.44      | 90.10      |
> | **50,000**      | 86.88              | 88.35      | 90.38      | 87.39                   | 88.75      | 90.43      |
> | **100,000**     | 86.62              | 88.16      | 89.93      | 87.02                   | 88.55      | 90.22      |
>
> The results indicate that MoEEdit is highly robust to the size of the preservation set. Using only 10,000 samples (10% of the original size) yields performance nearly identical to using 100,000 samples. This suggests that the expert activation manifolds are well-defined and can be captured efficiently, further reducing the pre-computation overhead of our method.

---

### Official Review · Reviewer_zj25 · 2025-10-29

**Soundness:** 2
**Presentation:** 3
**Contribution:** 3
**Rating:** 4
**Confidence:** 4

**Summary:**

The paper proposes MoEEdit, a method for routing-stable knowledge editing in MoE LLMs. This method reparameterizes expert updates through null-space projections for each expert, suppressing routing shifts. The method also introduces randomized block coordinate descent to efficiently implement expert updates, addressing the computational complexity issues caused by direct solving.

**Strengths:**

The paper's proposed MoEEdit effectively mitigates the routing shift problem in MoE LLMs editing while requiring only minimal computational cost. Experiments demonstrate that this method achieves better performance than other baseline methods across three metrics: Efficacy, Generalization, and Specificity.

**Weaknesses:**

- The paper does not explain how the v used in editing is obtained. As the output of the edited layer, v should also be one of the factors contributing to the routing shift problem, but the authors do not discuss this element.
- The paper lacks detailed descriptions of the experimental setup.
- The paper has some writing deficiencies, such as when the metric RS first appears in Section 4.1, it does not introduce that the full name of this metric is routing-similarity, which only appears later in Section 5.3.

**Questions:**

- How is the v used in editing obtained? What impacts does v have on the routing shift problem?
- What are the experimental settings used in the paper?

---

> ### Author Response · Authors · 2025-11-22
> **Response to Reviewer zj25**
>
> We sincerely thank the reviewer for recognizing the effectiveness of MoEEdit in mitigating routing shifts and its computational efficiency. We appreciate your constructive feedback regarding the definition of the target vector and experimental details. We have addressed these points in the revised manuscript, with the corresponding additions highlighted in red.
>
> ---
>
> > **W1&Q1: How is the $v$ used in editing obtained?**
>
> We apologize for the omission in the main text. The target vector $v$ is obtained following the standard "Locate-then-Edit" paradigm, which we have now detailed in the newly added Appendix B.1.
>
> Specifically, $v$ is the optimized hidden state that maximizes the probability of the new target fact (object $o^*$). It is computed by freezing the model parameters and optimizing a vector perturbation $\delta$ to the original output of the specific layer:
> $$
> v^\ast = v_{original} + \delta^\ast, \quad \text{where } \delta^\ast = \underset{\delta}{\arg\min} -\log \mathbb{P}(o^\ast \mid \text{do}(v \leftarrow v + \delta))
> $$
> This process translates the semantic edit target into a vector-space target $v$ used in Equation 5.
>
> > **W1&Q1: What impacts does $v$ have on the routing shift problem?**
>
> This is a crucial conceptual distinction. The target vector $v$ does not directly cause routing shift, but the parameter update required to achieve $v$ does.
>
> 1. **$v$ is an Objective, not an Input:** $v$ is a fixed optimization target derived from the teacher-forced forward pass. It is never fed into the model as an input during inference, nor does it propagate through the network.
> 2. **The Mechanism of Shift is $\Delta$:** To make the model output $v$, we must add a perturbation $\Delta$ to the expert weights ($W_{new} = W + \Delta$). It is this $\Delta$ that alters the hidden states $u$ for *subsequent* tokens. If $\Delta$ is unconstrained (as in standard FT), it perturbs $u$ in directions that the next layer's router is sensitive to, causing routing distribution shift.
> 3. **MoEEdit's Solution:** MoEEdit does not change $v$; instead, it constrains $\Delta$ via the null-space projector $P_n$ so that $\Delta$ produces $v$ for the target fact but remains "invisible" (zero effect) for general tokens, thereby preventing downstream routing shifts.
>
> ---
>
> > **W2&Q2: Experimental Setup Details**
>
> We agree that the initial description was too brief. We have added a comprehensive Appendix C (Experiments Setup) to the revised manuscript, which details model configurations, hardware, hyperparameters of MoEEdit and baselines.
>
> ---
>
> > **W3: Clarification of Terminology**
>
> Thank you for catching this oversight. We have corrected the manuscript to explicitly define RS (Routing Similarity) upon its first occurrence in Section 4.1. The text now reads:
>
> > _"We define the routing distribution shift... measured using the **Routing Similarity (RS)**, defined as the Jaccard similarity of Top-K expert sets..."_
>
> This ensures consistency with the usage in Section 5.3.

---

> ### Comment · Reviewer_zj25 · 2025-11-27
>
> Based on the author's responses, which have largely addressed my concerns, I now firmly believe this paper fully deserves acceptance. My position is based on the following two key points:
>   1. This is a direction that I have discussed with other knowledge editing researchers, and we consider it one of the worthwhile areas to explore.
>   2. Among recent knowledge editing papers I've reviewed, this work positions within the top 20% in terms of contribution and execution.
>
> Given that some reviewers may maintain negative assessments, I am raising my score 4 to 8 to strongly support its acceptance as a poster and ensure its contributions receive appropriate recognition.

---

> > ### Author Response · Authors · 2025-11-27
> >
> > We sincerely thank the reviewer for the prompt feedback and strong support. We are greatly encouraged by your recognition of our work's contribution and execution, as well as your shared perspective on the value of this research direction. Your positive assessment strengthens our confidence in the impact of MoEEdit.

---

### Official Review · Reviewer_5jmK · 2025-10-30

**Soundness:** 3
**Presentation:** 4
**Contribution:** 3
**Rating:** 6
**Confidence:** 3

**Summary:**

This paper introduces MoEEdit the first systematic framework for knowledge editing in Mixture-of-Experts or MoE LLMs. The authors identify a tripartite challenge for MoE editing computational cost expert coupling and most importantly routing distribution shift. They argue that naively editing MoE models perturbs layer outputs which in turn causes downstream routers to select different experts destabilizing the model. MoEEdit solves this by reparameterizing updates using per-expert null-space projections. This constrains the edits to not affect a preservation set which keeps router inputs invariant and prevents routing shift. To make this tractable they solve the block-structured optimization with an efficient randomized block coordinate descent BCD solver. Experiments on large MoE models show MoEEdit achieves state-of-the-art performance while being highly efficient and maintaining routing stability.

**Strengths:**

1. This is the first paper to properly formalize and tackle the knowledge editing problem for sparse MoE models.

2. The identification of routing distribution shift as the key failure mode for MoE editing is a novel and very important insight.

3. The proposed solution is elegant. The per-expert null-space projection directly targets the routing shift problem.

4. The BCD solver is a crucial component that makes the method practical. The paper clearly shows why a global one-shot solution is computationally infeasible and provides a scalable alternative.

**Weaknesses:**

1. The analysis focuses entirely on the routing shift in _subsequent_ layers. It's unclear how the edit affects routing for _subsequent tokens_ within the _same_ edited layer.

2. The null-space projection is critical but it relies on a preservation set $K_n^0$. The paper never explains how this set is collected how big it is or how sensitive the method is to its quality.

3. The BCD solver is efficient in terms of scaling with expert count $N$ but it's still an iterative process. Figure 4 shows it needs 6-10 passes. This makes it much slower per edit than one-shot dense editors like ROME. A wall-clock time comparison is needed.

4. The novelty versus AlphaEdit could be clearer. The paper should test what happens if you just apply a standard global AlphaEdit projection to the K-active experts. This would help isolate the benefit of the per-expert projection design.

**Questions:**

1. How do you build the preservation set $K_n^0$ for each expert? How many samples do you need for the projection matrix $P_n$ to be effective?

2. Your routing stability analysis looks at downstream layers. Does the edit also change the expert routing _within_ the edited layer for the tokens that follow the edit position?

3. The BCD solver needs multiple passes. Can you report the actual wall-clock time in seconds for a single edit batch? I'm trying to understand the real-world speed.

4. What happens if you just use a single global AlphaEdit-style projection on the active experts? I'm trying to figure out if the per-expert projection is the most critical part of your method.

---

> ### Author Response · Authors · 2025-11-22
> **Response to Reviewer 5jmK**
>
> We thank the reviewer for their insightful comments and for recognizing MoEEdit as the first systematic framework for MoE editing. We appreciate the validation of our core contributions, specifically the identification of routing distribution shift and the elegance of our null-space projection solution.
>
> Below, we address your questions regarding the preservation set, routing stability, computational efficiency, and ablation studies with additional experiments and data.
>
> ---
>
> > **W2&Q1: How do you build the preservation set $K_n^0$ for each expert?**
>
> To construct the preservation set $K_n^0$ for a specific expert $n$ at layer $l$, we adopt the following procedure:
>
> 1. Data Source: We sample sentences from a generic corpus (Wikipedia) to represent general knowledge.
> 2. Forward Pass & Filtering: We perform a forward pass on these samples. We specifically track the "fact token" (the last subject token) position.
> 3. Expert-Specific Collection: If expert $n$ is selected by the router (i.e., is in the Top-K) for a specific sample at the target token, we collect the post-gate key vector $k$.
> 4. Covariance Computation: These collected keys form the set $K_n^0$. We then compute the covariance matrix of $K_n^0$ to derive the null-space projector $P_n$, ensuring that updates to expert $n$ do not affect its behavior on these "preservation" inputs.
>
> ---
>
> > **W2&Q1: How many samples do you need for the projection matrix $P_n$ to be effective?**
>
> In our main experiments, we utilized a pool of 100,000 samples, which takes approximately 1.75 hours to process for projection matrix computation. To address your concern regarding sensitivity, we conducted an ablation study varying the sample size from 10,000 to 100,000.
>
> **Table R1: Effect of Preservation Set Size on Editing Performance (Qwen3-30B-A3B)**
>
> | Samples (N) | COUNTERFACT |       |       |       | ZsRE  |       |       |       |
> | ----------- | ----------- | ----- | ----- | ----- | ----- | ----- | ----- | ----- |
> |             | Eff.        | Loc.  | Spec. | Uti.  | Eff.  | Loc.  | Spec. | Uti.  |
> | 10,000      | 99.50       | 92.90 | 80.60 | 91.00 | 84.34 | 76.95 | 42.37 | 67.89 |
> | 20,000      | 99.40       | 93.90 | 80.99 | 91.43 | 84.56 | 77.45 | 42.27 | 68.09 |
> | 50,000      | 99.50       | 93.75 | 80.96 | 91.40 | 84.16 | 77.46 | 42.66 | 68.09 |
> | 100,000     | 99.30       | 94.10 | 80.97 | 91.46 | 84.47 | 78.01 | 42.82 | 68.43 |
>
> **Table R2: Effect of Preservation Set Size on Routing Stability (RS)**
>
> | Samples (N) | Editing Set RS |            |            | Preservation Set RS |            |            |
> | ----------- | -------------- | ---------- | ---------- | ------------------- | ---------- | ---------- |
> |             | Lay. 11-20     | Lay. 21-30 | Lay. 31-40 | Lay. 11-20          | Lay. 21-30 | Lay. 31-40 |
> | 10,000      | 87.04          | 88.42      | 89.38      | 87.41               | 88.60      | 89.53      |
> | 20,000      | 86.25          | 88.26      | 89.96      | 86.63               | 88.44      | 90.10      |
> | 50,000      | 86.88          | 88.35      | 90.38      | 87.39               | 88.75      | 90.43      |
> | 100,000     | 86.62          | 88.16      | 89.93      | 87.02               | 88.55      | 90.22      |
>
> As shown in Tables R1 and R2, performance saturates relatively quickly. Even with 10,000 samples, the method achieves high utility and stability. This demonstrates that MoEEdit is robust to the size of the preservation set and does not require excessive data to be effective.

---

> ### Author Response · Authors · 2025-11-22
> **Response to Reviewer 5jmK**
>
> >  **W1&Q2: Does the edit also change the expert routing within the edited layer for the tokens that follow the edit position?**
>
> We interpret the reviewer’s query regarding "tokens that follow the edit position" as a concern about autoregressive temporal propagation—specifically, how the edit affects the routing of subsequent tokens generated after the edited token.
>
> Based on this understanding, it is important to clarify the mechanism of the MoE layer. The router computes gating weights based on the input hidden state $u$ before the experts process that input.
>
> 1. **Current Token:** Since we freeze the router parameters and only update expert weights, the routing for the *current*token at the *edited* layer remains mathematically identical.
> 2. **Subsequent Tokens:** For autoregressive generation, the output of the edited layer changes the hidden states for subsequent layers and subsequent tokens. This creates a cascading effect.
>
> If we edit multiple layers (e.g., Layers 3-7), the routing of Layer 3 (the first edited layer) remains unchanged. However, the output of Layer 3 changes, which alters the input to Layer 4, thereby shifting the routing distribution of Layer 4, and so on.
>
> To quantify this, we applied edits to Qwen3-30B-A3B across layers [3, 4, 5, 6, 7] and measured the Routing Similarity (RS) shift within those specific layers for the edited samples.
>
> **Table R3: Layer-wise Routing Similarity (RS) in Edited Layers**
>
> | Layer Index | Layer 3 | Layer 4 | Layer 5 | Layer 6 | Layer 7 |
> | ----------- | ------- | ------- | ------- | ------- | ------- |
> | RS Score    | 1.00    | 0.9391  | 0.9295  | 0.8792  | 0.8719  |
>
> As expected, the first edited layer (Layer 3) has perfect stability (RS=1.0). The routing shift accumulates in subsequent layers (Layers 4-7) due to the altered hidden states flowing from the preceding edited experts. This confirms the cascading nature of routing shift and underscores the necessity of our null-space projection to minimize this drift.
>
> ----
>
> > **W3&Q3: Can you report the actual wall-clock time in seconds for a single edit batch?**
>
> We appreciate the reviewer raising the comparison with dense editors (e.g., ROME). It is worth noting that dense editors cannot be directly applied to MoE models due to the sparse activation and gating mechanisms. The most relevant comparison is the internal efficiency of our BCD solver, which addresses the dimensionality challenge of MoE experts.
>
> We measured the average wall-clock time for the BCD optimization process on a single batch (50 edits) when performing 1,000 sequential edits. The "BCD Opt. Time" refers to the time required to solve the linear system for the batch.
>
> **Table R4: Runtime of BCD Optimization (Per Batch of 50 Edits)**
>
> | **Dataset** | **Model** | **BCD Opt. Time** | **Time Per BCD Pass** |
> | --------------- | ------------- | ----------------- | --------------------- |
> | **ZsRE** | Qwen3-30B-A3B | 31.9s             | 1.6s                  |
> | **ZsRE** | GPT-OSS-20B   | 8.0s              | 0.8s                  |
> | **COUNTERFACT** | Qwen3-30B-A3B | 31.8s             | 1.6s                  |
> | **COUNTERFACT** | GPT-OSS-20B   | 7.9s              | 0.8s                  |
>
> The results demonstrate the computational efficiency of our proposed solver. For the large-scale Qwen3-30B model, the optimization for a batch of 50 edits is completed in approximately 32 seconds. With a latency of <2 seconds per optimization pass, the BCD solver scales effectively and facilitates practical deployment.

---

> ### Author Response · Authors · 2025-11-22
> **Response to Reviewer 5jmK**
>
> > **W4&Q4: What happens if you just use a single global AlphaEdit-style projection on the active experts? I'm trying to figure out if the per-expert projection is the most critical part of your method.**
>
> To investigate the necessity of our expert-aware design, we implemented the "Unified" baseline suggested by the reviewer. Instead of constructing individual preservation sets $K_n^0$ for each expert, we merged all preservation keys into a single global set and computed a Unified Projection Matrix applied to all updated experts (mimicking a global AlphaEdit-style constraint).
>
> Table R5 and Table R6 compare this Unified approach against our proposed Per-Expert MoEEdit on Qwen3-30B-A3B.
>
> **Table R5: Utility Comparison (Qwen3-30B-A3B)**
>
> | Method               | COUNTERFACT |       |       |       | ZsRE  |       |       |       |
> | -------------------- | ----------- | ----- | ----- | ----- | ----- | ----- | ----- | ----- |
> |                      | Eff.        | Loc.  | Spec. | Uti.  | Eff.  | Loc.  | Spec. | Uti.  |
> | MoEEdit (Per-Expert) | 99.30       | 94.10 | 80.97 | 91.46 | 84.47 | 78.01 | 42.82 | 68.43 |
> | MoEEdit (Unified)    | 99.50       | 95.30 | 78.55 | 91.12 | 84.50 | 78.34 | 42.60 | 68.48 |
>
> **Table R6: Routing Stability (RS) Comparison (Qwen3-30B-A3B)**
>
> | Method               | Editing Set RS  |            |            | Preservation Set RS  |            |            |
> | ------------------------ | ------------------- | ---------- | ---------- | ------------------------ | ---------- | ---------- |
> |                          | Lay. 11-20          | Lay. 21-30 | Lay. 31-40 | Lay. 11-20               | Lay. 21-30 | Lay. 31-40 |
> | **MoEEdit (Per-Expert)** | **86.62**           | **88.16**  | **89.93**  | **87.02**                | **88.55**  | **90.22**  |
> | **MoEEdit (Unified)**    | 83.83               | 85.71      | 87.48      | 84.36                    | 85.96      | 87.74      |
>
> The results demonstrate that the Per-Expert projection is critical for the method's success, particularly regarding stability:
>
> 1. Granular Control Preserves Routing: As shown in Table R6, the Unified projection suffers a consistent drop in Routing Stability (approx. 2-3 points across all layers). A global projection forces all experts to respect a "union" of constraints. This lacks the precision to distinguish which specific expert needs to be invariant to which specific feature. Consequently, the updates interfere with the delicate routing boundaries more frequently than our dedicated per-expert projectors.
> 2. Specific Knowledge Decoupling: In Table R5, while Efficacy remains high, the Specificity of the Unified approach drops (e.g., 80.97 $\to$ 78.55 on CounterFact). The per-expert design effectively decouples the optimization of different experts, ensuring that an edit to Expert A does not unnecessarily perturb the manifold of Expert B, thereby minimizing side effects.
>
> In summary, MoEEdit's per-expert design is not merely an implementation detail but a fundamental requirement to refine the optimization of each expert individually. This granularity effectively suppresses routing distribution shifts that a coarse, global projection cannot address.

---

### Official Review · Reviewer_MzWn · 2025-11-01

**Soundness:** 2
**Presentation:** 3
**Contribution:** 2
**Rating:** 4
**Confidence:** 4

**Summary:**

The paper targets knowledge editing (KE) for sparse Mixture‑of‑Experts (MoE) LLMs, arguing that naively adapting dense‑model editors leads to high compute, inter‑expert coupling, and routing distribution shift that destabilizes behavior in downstream layers. The authors  Introduce MoEEdit, which (i) reparameterizes per‑expert updates via a per‑expert null‑space projection constructed from preservation features to keep router inputs invariant (suppressing routing shift), and (ii) solves the resulting block‑structured problem with a randomized block coordinate descent (BCD) solver that updates only relevant experts.

**Strengths:**

- Clear identification of MoE‑specific failure modes (compute, inter‑expert coupling, routing drift) and a principled block‑structured edit formulation tailored to MoE.
- Practical, scalable solver: exact per‑expert ridge updates in a randomized BCD loop over active experts; clear normal equations and complexity discussion; convincing synthetic scaling vs. a global solver.
- Strong results on two MoE LLMs and two standard editing benchmarks.

**Weaknesses:**

- The central stabilization mechanism is a null‑space projection constructed from preservation features, which the paper itself positions as inspired by AlphaEdit’s null‑space constrained editing for dense models; the novelty is its per‑expert application in MoE and its connection to router invariance. While this is a reasonable extension, the claim to be “the first” routing‑stable framework may be too strong without comprehensive comparison to prior MoE‑specific KE or adaptor‑based lifelong editing.
- Missing comparison to MoE‑specific editing work (LEMoE) and absence of citation. The manuscript does not cite or compare to “LEMoE: Advanced Mixture of Experts Adaptor for Lifelong Model Editing of Large Language Models.” This omission weakens the novelty claim. Prior art already identified “inconsistent routing” as a key barrier in lifelong model editing and introduced a dedicated mechanism (KV anchor routing) to enhance routing consistency between training and inference. This directly contradicts MoEEdit’s novelty claim that it is “the first to formally identify routing-induced instability” as central. Both methods pursue the same core objective—maintaining routing stability during/after edits in MoE settings—via targeted constraints on the routing pathway. In addition, both target lifelong/sequential and batch editing regimes at scale with similar benchmarks and reporting conventions.

**Questions:**

Please see Weaknesses.

---

> ### Author Response · Authors · 2025-11-22
> **Response to Reviewer MzWn**
>
> We sincerely thank the reviewer for their constructive feedback and for bringing LEMoE [1] to our attention. We appreciate the opportunity to clarify the positioning of our work.
>
> > **Q1: The manuscript misses a comparison to LEMoE. Since both methods address "routing stability" in MoE settings, how does MoEEdit differ from LEMoE in terms of architecture and problem definition?**
>
> While both works utilize the term "MoE" and discuss "routing," they address fundamentally different problems in distinct architectural settings. Below, we detail why MoEEdit remains novel and why the "routing stability" in MoEEdit is distinct from that in LEMoE. We have incorporated LEMoE into the related work section of our revised manuscript and highlighted the corresponding additions in blue to clarify these distinctions.
>
> **1. Fundamental Architectural Distinction: Dense vs. Sparse Models**
> The primary distinction lies in the target architecture and the editing paradigm:
>
> * **LEMoE is a Parameter-Preserving Adaptor for Dense Models:** LEMoE does not edit Mixture-of-Experts models. Instead, it edits dense models (specifically LLaMA-2-7B and Mistral-7B) by inserting an external "MoE Adaptor" module. It freezes the original model parameters and adds new, parallel FFN experts to handle edits. It is effectively a dynamic LoRA/Adapter approach inspired by MoE structures.
> * **MoEEdit is a Parameter-Modifying Editor for Sparse MoE Models:** Our work targets native Sparse MoE LLMs (e.g., Qwen3-MoE, GPT-OSS). We operate directly on the pre-existing experts of the base model. We do not add external modules; rather, we modify the weights of specific experts in-place.
>
> **2. Distinct Definitions of "Routing Stability"**
> The reviewer correctly notes that both papers discuss routing, but the definitions and mechanisms are orthogonal:
>
> * **LEMoE's Routing Consistency (Internal Adaptor Stability):** LEMoE's challenge is ensuring that its own added adaptor router selects the same added expert during training and inference. Their solution (KV Anchor Routing) aligns the router's selection consistency for the inserted module. Crucially, the base model's parameters are frozen, so the signal flow through the rest of the network remains static.
> * **MoEEdit's Routing Stability (Downstream Distribution Shift):** MoEEdit tackles a deeper, systemic issue in multi-layer sparse networks. When we modify parameters in Layer $L$ of a sparse model, the output representations shift. This shift propagates to Layer $L+1$, changing the input to the next router. This causes Routing Distribution Shift, where downstream layers activate the "wrong" experts for unedited tokens.
>
> > **Q2: Given the prior existence of LEMoE, is the claim that MoEEdit is "the first" routing-stable framework too strong?**
>
> We agree with the reviewer that precise language is vital. We will revise our claim to explicitly state:
>
> > "*MoEEdit is the first routing-stable framework for **parameter-modifying knowledge editing in native Sparse Mixture-of-Experts LLMs**.*"
>
> We add a discussion of LEMoE in our "Related Work" section. And we credit it for identifying routing consistency within adaptors, while contrasting it with our focus on mitigating cascading routing drift in sparse architectures. We believe this clarification demonstrates that MoEEdit addresses a novel, unaddressed challenge in the regime of Sparse LLMs that LEMoE does not cover.
>
>
> [1] Renzhi Wang and Piji Li. LEMoE: Advanced Mixture of Experts Adaptor for Lifelong Model Editing of Large Language Models. EMNLP 2024

---

### Author Response · Authors · 2025-12-03
**Rebuttal Summary by the Authors**

We sincerely thank the Area Chair and reviewers for their time and constructive feedback. During the rebuttal phase, we provided detailed responses and conducted extensive additional experiments to address all raised concerns.

**Summary of Improvements and Clarifications**

Guided by reviewer feedback, we have strengthened the manuscript in three key areas:

**1. Positioning against Prior Work (Reviewer MzWn)**
We clarified the fundamental distinction between MoEEdit and LEMoE. We emphasized that while LEMoE acts as a parameter-preserving adaptor for dense models, MoEEdit is the first framework to tackle parameter-modifying editing for native sparse MoE models. We have revised our claims to be precise and added detailed discussions in the Related Work section.

**2. Efficiency and Scalability Verification (Reviewers 5jmK, fg5e)**
To address concerns regarding computational costs and large-scale stability, we added:
* *Wall-clock runtime analysis:* Demonstrating our BCD solver completes a batch of 50 edits in ~32 seconds on Qwen3-30B, validating practical efficiency.
* *Extended stress testing:* We scaled sequential editing experiments from 1,000 to 5,000 edits. Results show that MoEEdit maintains high efficacy (>96%) and Routing Stability (>80%) even under this extreme load, confirming the robustness of our null-space projection.

**3. Mechanism Validation and Ablations (Reviewers 5jmK, zj25)**
* *Per-Expert vs. Global Projection:* We performed new ablations (Table R5/R6 in rebuttal) proving that our per-expert design significantly outperforms a global (AlphaEdit-style) projection in maintaining routing stability.
* *Preservation Set Sensitivity:* We demonstrated that performance is robust even when reducing the preservation set size from 100k to 10k samples.
* *Theoretical Clarity:* We added Appendix B.1 and C to explicitly define the derivation of the target vector $v$ and detail all experimental hyperparameters. **Solving these concerns led Reviewer zj25 to raise their evaluation from 4 to 8.**

We believe the added experiments and clarifications have fully resolved the reviewers' initial concerns, establishing MoEEdit as a robust baseline for editing sparse LLMs.

---

### Meta-Review · Area_Chair_rexh · 2026-01-06

**Summary:**

This paper presents MoEEdit, a framework for knowledge editing in sparse Mixture-of-Experts (MoE) large language models. Reviewers generally recognize the novelty and value of addressing the routing distribution shift problem in MoE LLMs, and they acknowledge that the proposed per-expert null-space projection with a BCD solver is an elegant and practical solution. Experimental results across multiple models and benchmarks further support the method’s efficacy and robustness.

Reviewers’ concerns primarily revolve around three areas:

Novelty and positioning: Reviewer MzWn noted that the paper’s claim to be “first” in routing-stable MoE editing could be overstated without comparison to prior work like LEMoE. This was addressed by clarifying the distinction between parameter-modifying edits in native sparse MoEs versus adaptor-based edits in dense models.

Experimental clarity and details: Several reviewers (5jmK, fg5e, zj25) requested more information on the preservation set, runtimes, ablation studies, and large-scale sequential edits. The authors have supplemented this with additional experiments, and detailed appendices.

Minor writing and presentation issues: Some terminology (e.g., RS) and experimental setup descriptions were initially unclear, which have now been clarified.

Overall, while the paper is incremental and inspired by prior work such as AlphaEdit, it extends these ideas meaningfully to MoE models, addressing novel challenges and providing practical, scalable solutions. The work is clearly written, and experimentally validated, and the reviewers’ concerns have been largely addressed.

**Reviewer Concerns:**

The rebuttal effectively addresses most of the reviewers’ concerns. Key points such as the distinction from prior work (LEMoE), the construction and robustness of the preservation set, computational efficiency, and clarification of target vectors and terminology have all been satisfactorily explained or supplemented with additional experiments and appendices. Remaining concerns are minor and mainly pertain to further experimental extensions that do not affect the overall validity or contribution of the work.

**Reviewer Scores:**

MzWn: Initially rated marginally below threshold due to novelty and related work concerns. With the clarification distinguishing MoEEdit from LEMoE and the refined “first” claim, the reviewer would likely raise their score to the acceptance threshold.

5jmK: Already positive and supportive, with minor concerns about preservation set details and ablation studies. With the additional experiments and clarifications provided, the score would likely remain the same or slightly increase.

fg5e: Slightly above threshold, concerned about efficiency and extended edits. The rebuttal’s runtime tables and large-scale edit experiments would likely increase confidence and lead to a higher score.

zj25: With the issues clarified in the rebuttal, the reviewer would likely raise their score toward acceptance.

---

### Decision · Program_Chairs · 2026-01-26

Accept (Poster)